# Mouse skeletal muscle satellite cells co-opt the tenogenic gene *Scleraxis* to instruct regeneration

Yun Bai[1], Tyler Harvey[1], Colin Bilyou[1], Minjie Hu[2]*, Chen-Ming Fan[1]*

[1]Department of Embryology, Carnegie Institution for Science, Baltimore, United States; [2]College of Life Sciences, Zhejiang University, Hangzhou, China

## eLife Assessment

This manuscript presents **important** finding regarding the regulation of a key stem cell population, namely muscle stem cells (or "satellite cells"). The evidence presented is **convincing** that Scx, a marker for tendon, is expressed in some myogenic cells and is essential for adult muscle regeneration.

**\*For correspondence:**
minjie-hu@zju.edu.cn (MH);
fan@carnegiescience.edu (C-MF)

**Competing interest:** The authors declare that no competing interests exist.

**Abstract** Skeletal muscles connect bones and tendons for locomotion and posture. Understanding the regenerative processes of muscle, bone, and tendon is of importance to basic research and clinical applications. Despite their interconnections, distinct transcription factors have been reported to orchestrate each tissue's developmental and regenerative processes. Here, using adult mouse skeletal muscles, we show that *Scx* expression is not detectable in adult muscle stem cells (also known as satellite cells, SCs) during quiescence. *Scx* expression begins in activated SCs and continues throughout regenerative myogenesis after injury. By SC-specific *Scx* gene inactivation (*Scx* cKO), we show that *Scx* function is required for SC expansion/renewal and robust new myofiber formation after injury. We combined single-cell RNA sequencing and CUT&RUN to identify direct Scx target genes during muscle regeneration. These target genes help explain the muscle regeneration defects of *Scx* cKO and are not overlapping with *Scx*-target genes identified in tendon development. Together with a recent finding of a subpopulation of *Scx*-expressing connective tissue fibroblasts with myogenic potential during early embryogenesis, we propose that regenerative and developmental myogenesis co-opt the *Scx* gene via different mechanisms.

## Introduction

Regeneration of adult skeletal muscles following injury is initiated by the activation and proliferation of satellite cells (SCs). After extensive proliferation, progenitors undergo differentiation and fusion with each other or existing myofibers to recreate functional muscle tissue (*Yin et al., 2013*; *Liu et al., 2014*; *Fukada et al., 2022*). The intrinsic and extrinsic factors regulating myogenesis have been extensively investigated. The key transcription factors governing this process are largely the same as those deployed during embryogenesis, including paired-homeodomain proteins Pax3 and Pax7, basic helix-loop-helix (bHLH) myogenic regulatory factors (MRFs), such as Myf5 and Myod1, and the myocyte enhancer factor 2 (MEF2) family; however, their relative contribution or redundancy varies between the two processes (*Hernández-Hernández et al., 2017*). To date, resident Pax7[+] SCs are recognized as the major source of muscle stem cells in adult limb muscles. None of these myogenic transcription factors are known to participate in tendon development or regeneration.

Both muscle and tendon progenitors reside in the somite during embryonic development but are located in different compartments. *Pax3* and *Pax7* are expressed in the dermomyotome, which gives rise to the myotome expressing *Myf5* and/or *Myod1*. The syndetome, on the other hand, is defined by the expression of the earliest tenogenic progenitor marker *Scx* and gives rise to tendon and ligament (*Brent et al., 2003*). Like Myf5 and Myod1, Scx is a bHLH transcription factor, and they all bind to a DNA sequence motif called the E-box (*Cserjesi et al., 1995*). *Scx* expression persists in mature tenocytes, ligaments, and connective tissue fibroblasts (CT) (*Murchison et al., 2007*). *Scx* mutant mice have poorly developed tendons with drastically reduced expression of tendon matrix genes (*Murchison et al., 2007*; *Yoshimoto et al., 2017*; *Shukunami et al., 2018*). In adult tendon regeneration, the Tppp3+Pdgfra+ tendon stem cell population turns on *Scx* for tendon regeneration (*Harvey et al., 2019*). Lastly, the *Scx* function is required in post-natal tendon growth and regeneration (*Howell et al., 2017*; *Sakabe et al., 2018*; *Gumucio et al., 2020*; *Korcari et al., 2022*).

Intriguingly, lineage tracing using a constitutive *Scx^Cre* in mouse embryos found descendant cells in cartilage, tendon, ligament, muscle, and muscle interstitial CT (*Yoshimoto et al., 2017*; *Esteves de Lima et al., 2021*; *Ono et al., 2023*), suggesting that *Scx* is expressed either in several distinct musculoskeletal subpopulations or in a common progenitor that gives rise to different fates. Ablation of embryonic Scx+ cells causes a change in muscle bundling (*Ono et al., 2023*), presumably due to the loss of instructive cues from the tendon (or CT) to form proper muscle pattern (*Kardon, 1998*). In adult muscles, Hic1+ quiescent mesenchymal progenitors (MPs) give rise to Scx+ cells in the muscle interstitial compartment, and ablation of Hic1+ cells negatively impacts muscle regeneration (*Scott et al., 2019*). Muscle interstitial Scx+ cells engrafted into the muscle contribute only to extracellular matrix remodeling (*Giordani et al., 2019*). A survey of muscle interstitial CT assigned a sub-population of cells expressing tendon markers (including *Scx*) as paramysial cells - cells lining next to the perimysium that wraps around muscle fascicles (*Muhl et al., 2020*). Furthermore, Strenzke and colleagues showed that secretome from Scx overexpressed cells could significantly increase myoblast fusion and metabolic activity in vitro (*Strenzke et al., 2020*). Collectively, these data indicate that while some embryonic Scx+ cells can incorporate into myofibers, adult Scx+ cells contribute to skeletal muscle architecture and repair/regenerative processes in a paracrine manner.

Serendipitously, in the ScxGFP transgenic mouse Tg-ScxGFP (*Pryce et al., 2007*), we observed GFP fluorescence in SCs and regenerating myofibers after injury. We conducted a series of experiments to show that endogenous *Scx* is expressed in activated SC after injury. We show that *Scx* is functionally relevant in muscle regeneration by inactivating *Scx* in *Pax7*++ (*Scx* cKO). We employed single-cell RNA-sequencing (scRNA-seq) and CUT&RUN to define Scx's target genes during muscle differentiation and fusion. Down-regulation of Scx's target genes, such as *Mef2a*, *Cflar*, *Capn2*, and *Myh9* explains the regenerative defects of *Scx* cKO mice. In contrast to adult Scx+ muscle CT and embryonic muscle-forming Scx+ cells, our findings reveal a previously unappreciated role of *Scx* in adult Pax7+ SCs.

## Results

### ScxGFP transgene is expressed in the regenerative myogenic lineage

When we analyzed tibialis anterior (TA) muscles of the Tg-ScxGFP (ScxGFP) mice, scattered GFP+ cells were found in the interstitial space, but not in quiescent Pax7+ SCs nor in myofibers (*Figure 1—figure supplement 1A*). Unexpectedly, we found GFP signal in injured muscles. In cardiotoxin (CTX), injured TA muscles of ScxGFP mice at 5 days post-injury (dpi) (*Figure 1A*), we found GFP colocalized with Pax7+ SCs (*Figure 1B*). GFP also overlaps with committed myogenic progenitor marker MyoD1, myocyte marker Myogenin (Myog), and myosin heavy chain (MHC) in terminally differentiated myofibers (*Figure 1C*; *Figure 1—figure supplement 1B–D* for split channels). When we analyzed muscles administered with 5-ethynyl-2'-deoxyuridine (EdU) for 5 days after CTX injury (*Figure 1A*), GFP was found to colocalize with proliferated (EdU+) Pax7+ cells (*Figure 1D*). Thus, ScxGFP is expressed, albeit at varying levels, in the myogenic lineage during the regenerative process.

We next determined ScxGFP expression in cultured SCs. For this, we employed a four-surface marker fluorescent activated cell sorting (FACS) scheme (Sca1-CD31-CD45-Vcam1+) (*Liu et al., 2015*) to purify SCs from hindlimb muscles of ScxGFP mice (*Figure 1E*; *Figure 1—figure supplement 1E, F*); ~98% of isolated cells were Pax7+ (*Figure 1—figure supplement 1G, H*). While Pax7 was detected

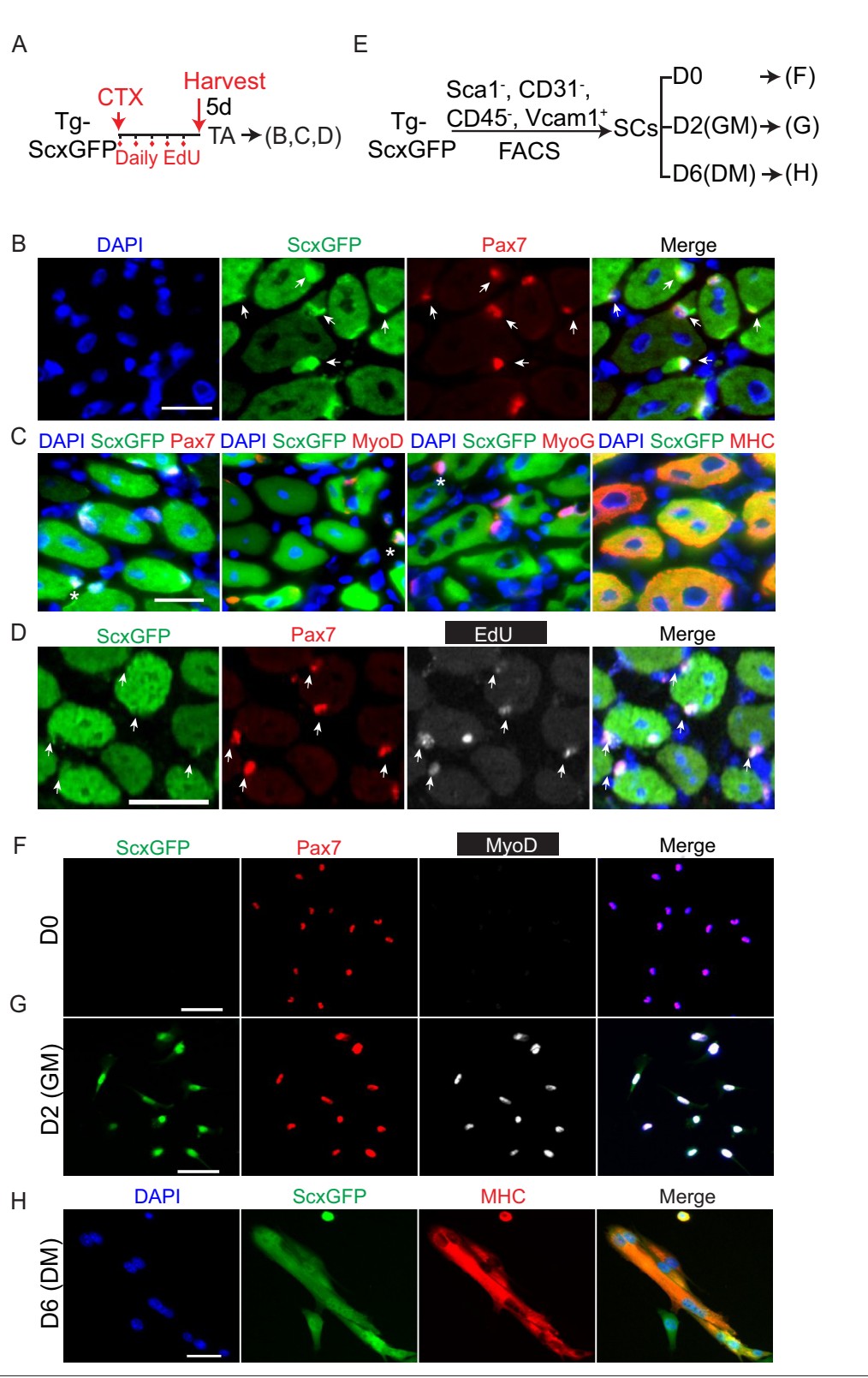

**Figure 1.** Adult regenerative myogenic cells express the transgene ScxGFP. (**A**) Experimental scheme for data in (**B–D**). Tg-ScxGFP (ScxGFP) mice were injured by cardiotoxin (CTX) to the tibialis anterior (TA) muscle, followed by daily 5-ethynyl-2′-deoxyuridine (EdU) administration for 5 days (5d), and their TA muscles were harvested for analysis at 5 days post-injury (dpi). (**B**) Muscle samples obtained in (**A**) were sectioned and stained with Pax7 and

*Figure 1 continued*

GFP (for ScxGFP expression) antibody. Arrows indicate Pax7 and ScxGFP double-positive cells; 97.77% Pax7$^+$ SCs were ScxGFP$^+$ (N=4 mice; n=1274 cells). (**C**) Muscle samples obtained in (**A**) were sectioned and stained in pairs of GFP/Pax7, GFP/MyoD1, GFP/MyoG, GFP/MYH (N=4 mice). Asterisks indicate cells double-positive for ScxGFP and each respective myogenic marker. All myofibers are GFP and myosin heavy chain (MHC) double positive, thus without additional labeling. (**D**) Muscle samples obtained in (**A**) were sectioned and stained for Pax7 and GFP, followed by EdU reaction (N=4 mice). Arrows indicate Pax7, ScxGFP, and EdU triple-positive satellite cells (SCs). (**E**) Experimental scheme of SC isolation from Tg-ScxGFP hindlimb muscles using four surface markers (CD31$^-$, CD45$^-$, Sca1$^-$, Vcam1$^+$) by FACS. Isolated SCs were assayed immediately after isolation (D0; data in **F**), after culture in growth media for 2 days (D2(GM); data in (**G**)), or after cultured for 4 days in GM followed by 2 days in differentiation media (DM) (D6(DM); data in **H**). F-G. D0 (**F**) and D2 cultured (**G**) SCs obtained in (**E**) were stained for GFP (i.e. ScxGFP), Pax7, and MyoD. At D0, no Pax7$^+$ cells were GFP$^+$, or MyoD$^+$. At D2, 95.3% of Pax7$^+$ cells were GFP$^+$, whereas 99.2% of MyoD$^+$ were GFP$^+$. (N=3 mice; n=1805 cells at D0; n=1332 cells at D2). (**H**) D6(DM) cells obtained in (**E**) were stained for GFP (i.e. ScxGFP) and MHC. 94.58% MHC$^+$ were GFP$^+$. (N=3 mice; n=1539 nuclei in MHC$^+$ domain examined). Nuclei were stained with DAPI (blue); Scale bars = 20 μm.

The online version of this article includes the following figure supplement(s) for figure 1:

**Figure supplement 1.** ScxGFP transgene is expressed in tendon, regenerative muscle, and cultured myoblast.

---

in the SC immediately after FACS isolation, neither MyoD nor GFP was detected (*Figure 1F*). After 2 days in culture, most cells were Pax7, MyoD, and GFP triple positive (*Figure 1G*). After switching to differentiation media for 2 days, GFP signal persisted in MHC$^+$ myotubes (*Figure 1H*). We, therefore, conclude that ScxGFP expression is initiated after SC becomes activated and continues into differentiated myofibers in vivo and in vitro.

## Endogenous *Scx* is expressed in activated SCs

To ensure that the ScxGFP expression observed in adult regenerative myogenesis is not caused by mis-expression due to transgene insertion site, we utilized *Scx$^{CreERT2}$* for tamoxifen (TMX) inducible lineage tracing with a tdTomato (tdT) reporter (Rosa26$^{fs-TdT}$)(*Madisen et al., 2010*). Two experimental groups with different TMX and injury regimens were designed (*Figure 2A*): (1) TMX-induced marking before injury, and (2) TMX-induced marking after injury; muscles were harvested at 14 dpi for analysis. Mice treated with TMX before injury showed little to no tdT$^+$Pax7$^+$ SCs or tdT$^+$ regenerative muscle fibers (identified by centrally located nuclei). By contrast, mice treated with TMX after injury showed ~30% of Pax7$^+$ SCs and all regenerated myofibers as tdT$^+$ at 14 dpi (*Figure 2C and D*). Pax7$^+$ SC densities were not different between these two groups (*Figure 2—figure supplement 1A*). These data extend the ScxGFP results in that (1) marked interstitial Scx$^+$ cells prior to injury do not possess myogenic potential, (2) endogenous *Scx* is expressed in activated SCs for regenerative myogenic lineage-marking, and (3) lineage-marked *Scx$^+$* SCs are capable of renewal as Pax7$^+$ SCs at 14 dpi. Examination of two published SC bulk RNA-seq data confirmed *Scx* expression in SCs isolated from wild-type (*Li et al., 2019*) and *mdx* mice (*Madaro et al., 2019*; *Figure 2—figure supplement 1B*). Re-analysis of published scRNA-seq data sets of regenerative myogenic cells also uncovered a widespread *Scx* expression at 2 dpi (*De Micheli et al., 2020*) and 2.5 dpi (*Dell'Orso et al., 2019*; *Figure 2E*; and more below), but not in freshly isolated SCs from uninjured muscles (*Dell'Orso et al., 2019*; *Figure 2—figure supplement 1C*). As those prior studies did not focus on *Scx*, its expression might not have been paid attention to. By contrast, our serendipitous finding from ScxGFP mice has led us to document *Scx* expression in activated SCs and regenerative myogenic cells in vivo and in vitro.

## *Scx* is required for adult skeletal muscle regeneration

To determine whether *Scx* plays a direct role in the myogenic lineage during regeneration, we combined floxed *Scx* (*Scx$^f$*)(*Murchison et al., 2007*) and *Pax7$^{Cre-ERT2}$* (Pax7$^{CE}$) (*Lepper et al., 2009*) to generate Scx cKO mice for TMX-inducible gene inactivation (*Figure 3—figure supplement 1A*); loxP sites flank the first exon of Scx (*Figure 3—figure supplement 1B*). Either tdT or YFP (R$^{YFP}$) reporter (specified in figures and legends) was included for cell marking. Highly efficient and selective removal of exon 1 was determined using genomic DNA samples of FACS-isolated control and Scx cKO SCs (*Figure 3—figure supplement 1A–D*).

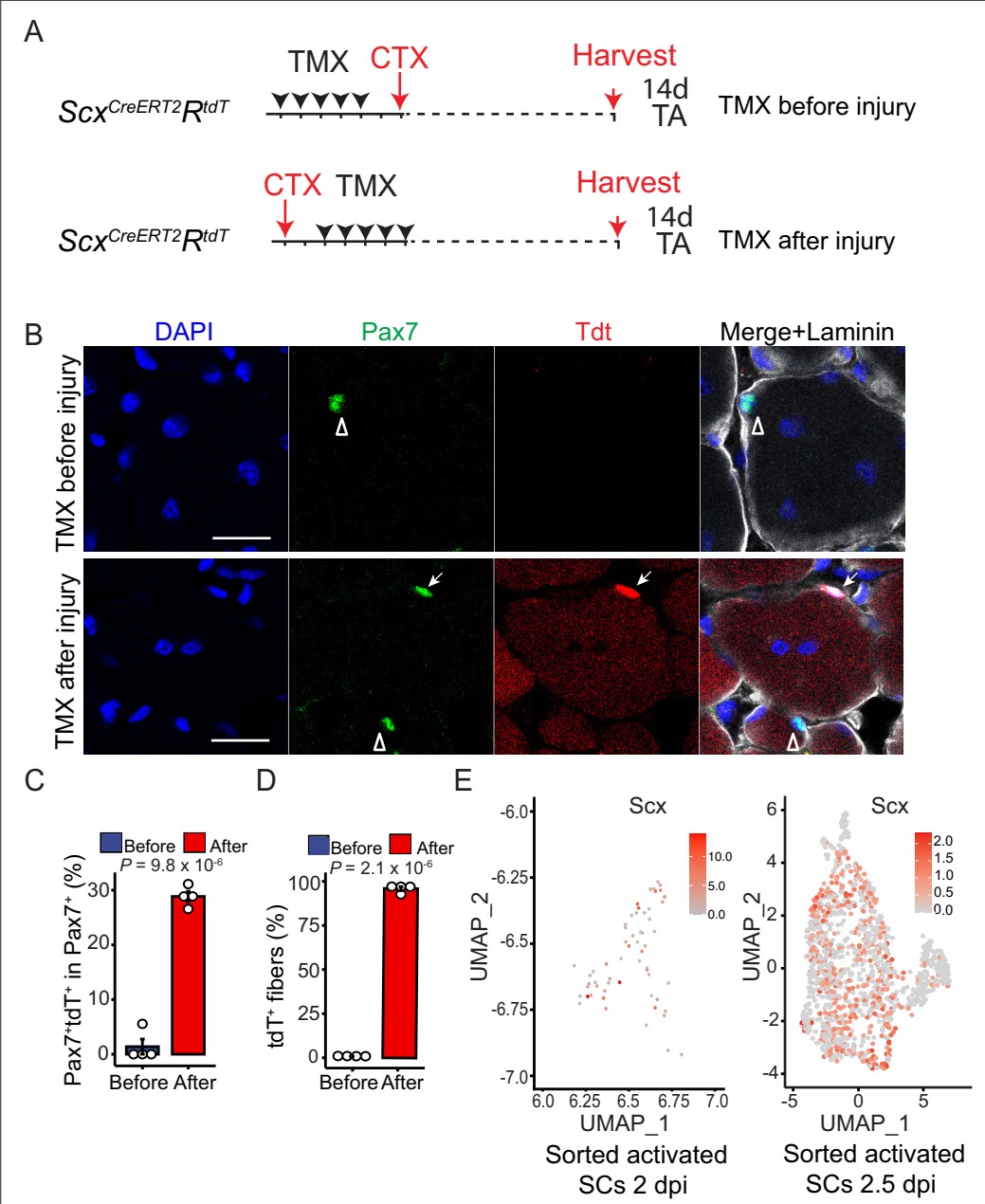

**Figure 2.** Endogenous *Scx* is expressed by activated but not quiescent satellite cells (SCs). (**A**) Experimental design for *Scx^CreERT2*-mediated inducible lineage tracing with the *R^tdT* reporter. The two experimental groups are: (1) Tamoxifen (TMX) administered before injury for 5 days (TMX before injury) and (2) TMX administered after injury for 5 days (TMX after injury). Tibialis anterior (TA) muscles in both groups were harvested at 14 days post-injury. (**B**) TA muscles from experiment groups in (**A**) were stained with Pax7 (green) and Laminin (white) and visualized with tdT (no staining). Open arrowheads indicate Pax7+ SCs; arrows, Pax7+tdT+ SCs. (**C**) Percentages of Pax7+tdT+ SCs in Pax7+ SCs examined, from data in **B**. (N=4 mice per group; n=191 (Before, TMX before injury) and 256 (After, TMX after injury) Pax7+ cells). (**D**) Percentages of tdT+ myofibers in regenerated muscle fibers (with centrally located nuclei), from data in **B**. (N=4 mice; n=1956 (Before, TMX before injury) and n=2024 (After, TMX after injury) regenerated myofibers). (**E**) Re-analyses for *Scx* expression in two published scRNA-seq data sets of activated myogenic cells at 2 dpi and 2.5 dpi (*De Micheli et al., 2020*; *Dell'Orso et al., 2019*), displayed by UMAP; colored keys to expression levels are included correspondingly. Nuclei were stained with DAPI; Scale bar = 20 μm. Data are presented with mean ± s.d.; *p*-values are indicated. (**C, D**) Unpaired two-tailed Student's *t*-test were applied.

The online version of this article includes the following figure supplement(s) for figure 2:

**Figure supplement 1.** Pax7+ and Scx expression in published data set.

## *Scx* cKO mice have muscle regeneration defects

Next, we injured control and *Scx* cKO mice with CTX and compared their regeneration at 5 and 14 dpi (*Figure 3A*); samples shown in *Figure 3* carried the tdT reporter. At 5 dpi, *Scx* cKO regenerating myofibers were significantly smaller than those in control mice (*Figure 3B and C*). Similar results were obtained in mice carrying the YFP reporter (*Figure 3—figure supplement 1J, K*). At 14 dpi, regenerated myofibers in *Scx* cKO mice were still considerably smaller than those of the control (*Figure 3D and E*). Thus, the *Scx* function is needed in the Pax7$^+$ SC lineage for robust regeneration of muscle fibers.

## *Scx* cKO mice show reduced SC proliferation and renewal

Considering that *Scx* expression is initiated in activated SCs but not in quiescent SCs, and *Scx* cKO mice have smaller regenerative myofibers, it stands to reason that *Scx* plays a role in their proliferation. At 5 dpi, we noted a ~ sevenfold reduction in Pax7$^+$ SCs in the *Scx* cKO samples (*Figure 3F*, *Figure 3—figure supplement 1E, F*). To show the proliferation defect, we administered EdU and assessed the cumulative proliferation index over the first 5 days of injury (*Figure 3A*, top panel). Compared to the control, the fraction of Pax7$^+$ *Scx* cKO SCs that incorporated EdU (i.e. EdU$^+$Pax7$^+$) was reduced by ~ fourfold (*Figure 3G and H*). We did not observe appreciable levels of programmed cell death (PCD) in control and *Scx* cKO at 5 dpi using an anti-cleaved Caspase 3 antibody. We also quantified Pax7$^+$ SCs number at 14 dpi (*Figure 3A*, bottom panel) and found a ~ fivefold reduction of renewed SCs in the *Scx* cKO group (*Figure 3I*, *Figure 3—figure supplement 1H, I*). Thus, Scx is autonomously required for SC proliferation and renewal following injury.

When we examined laminin (i.e. basement membrane) and MHC in control 5 dpi samples, we found that the laminin boundary juxtaposed the regenerative myofiber surface (*Figure 3—figure supplement 1F*). As expected, the smaller *Scx* cKO MHC$^+$ fibers did not fill out to the laminin outlines (*Figure 3—figure supplement 1G*). At early injury time points, the laminin pattern represents leftover basement membranes of dead myofibers (caused by injury), i.e., the ghost fiber (*Webster et al., 2016*). Ghost fibers are thought to be replaced by basement membranes produced by regenerated fibers over time. At 14 dpi, regenerated myofibers in both control and *Scx* cKO were tightly surrounded by laminin despite their difference in size (*Figure 3—figure supplement 1H, I*), suggesting that *Scx* cKO regenerated myofibers are capable of making their own basement membranes.

## Scx is needed for robust proliferation of SC in culture

To examine Scx function in the SC without interactions with other cell types in the injured/regenerative environment, we turned to in vitro assays using purified SCs (*Figure 4A*). To simplify FACS isolation of SC, we utilized either the tdT or the YFP fluorescent reporter (*Figure 4—figure supplement 1A, B*). SC purity was assessed by staining for Pax7 immediately after FACS. We were surprised that YFP-marked SCs (YFP-SCs) exhibited higher purity than tdT-marked SCs (*Figure 4B*). Indeed, Murach and colleagues have reported exosomal transfer of tdT mRNA from lineage-marked Pax7$^+$ cells to several other cell types (*Murach et al., 2021*). Their finding helps explain lower purity of Pax7$^+$ cells by tdT marking in our hands. We suggest that high levels of tdT mRNA produced by the strong CAG promoter/enhancer lead to more tdT$^+$ non-SCs by exosomal transfer, compared to the low-moderate levels of YFP reporter mRNA produced by the Rosa26 promoter.

As such, we opted to use YFP-marked control and *Scx* cKO SCs in subsequent studies for higher SC purity. Of note, *Scx* cKO with YFP reporter had similar regenerative defects as that with tdT reporter (*Figure 3—figure supplement 1J, K*). By EdU incorporation assay, we found that cultured *Scx* cKO SCs displayed reduced proliferation indices at days 2, 3, and 4. Curiously, the number of *Scx* cKO cells per imaged area (i.e. cell density) barely increased during this time course, despite EdU incorporation (*Figure 4C–E*). We next carried out live imaging (*Figure 4—figure supplement 1C*) to document the behavior of control and *Scx* cKO SCs. Consistent with EdU incorporation, control SCs showed a faster increase in cell number/density than *Scx* cKO SCs (*Figure 4—figure supplement 1D, E*). *Scx* cKO cells also showed a slightly reduced cell mobility (*Figure 4—figure supplement 1F*). As we did not observe appreciable levels of PCD using anti-cleaved Caspase 3 at 5 dpi, we assessed PCD by another assay (TUNEL) at an earlier time point. Yet, we still failed to detect appreciable TUNEL$^+$YFP$^+$ myogenic cells in either control or *Scx* CKO muscles at 3 dpi (*Figure 4—figure supplement 1I*). Intriguingly, we did observe cell loss during live imaging (*Figure 4—figure supplement 1G, H*): More *Scx* cKO cells

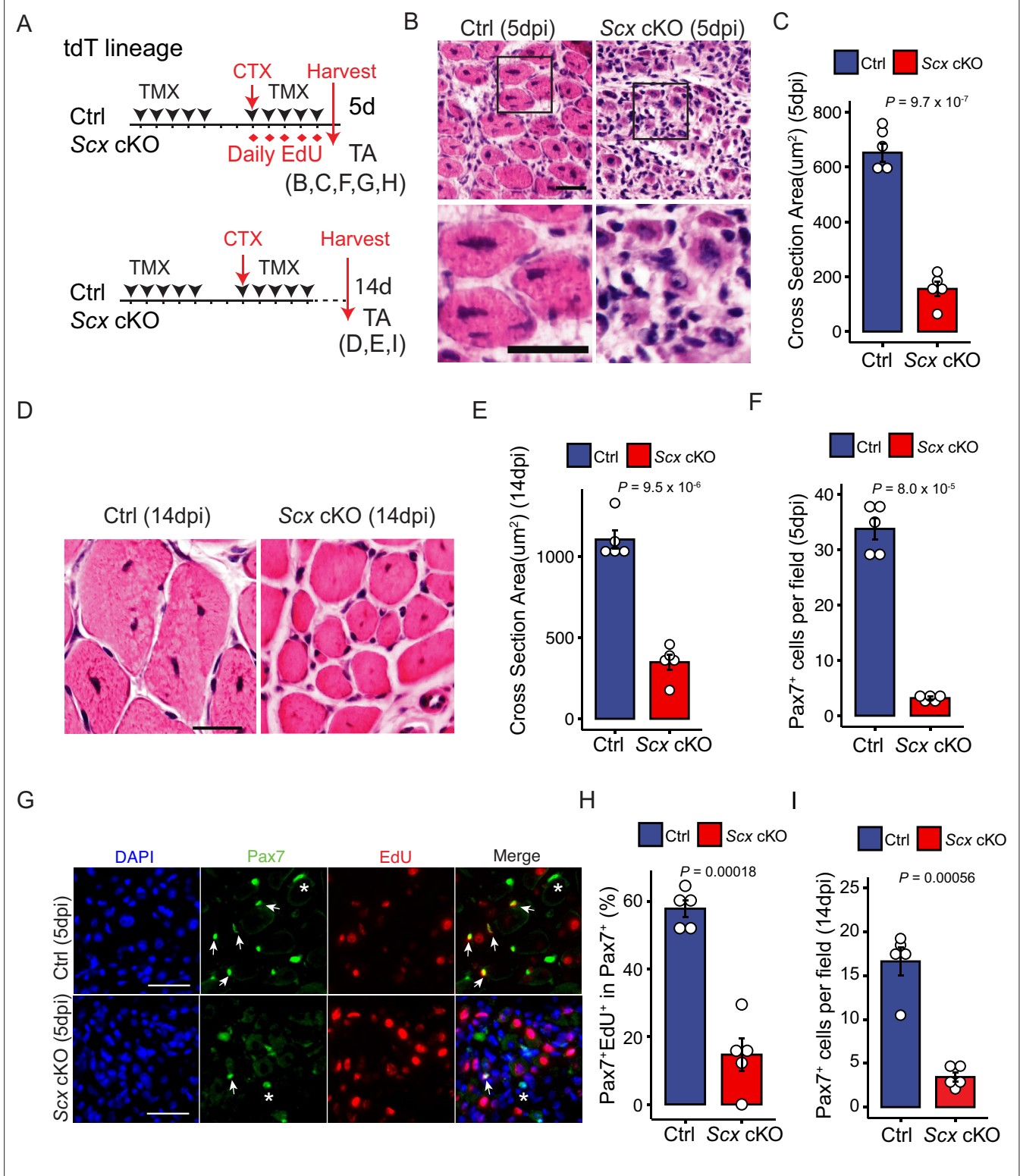

**Figure 3.** Efficient muscle regeneration requires *Scx* function. (**A**) Experimental designs to compare phenotypes of control (Ctrl) and *Scx* cKO mice. The $R^{tdT}$ reporter was included (tdT lineage; see ***Figure 3—figure supplement 1A*** for genotypes). Tamoxifen (TMX) was administered before and after the cardiotoxin (CTX)-induced injury to maximize gene inactivation. Tibialis anterior (TA) muscles were harvested at 5 days or 14 days after injury. (**B, C**) (**B**) Ctrl and *Scx* cKO TA muscles at 5 days post-injury were sectioned and stained with hematoxylin and eosin (H&E) at low (top) and high (bottom) magnifications. (**C**) Histogram of regenerated muscle fiber cross-sectional area from data in (**B**). (N=5 mice per group). (**D, E**) (**D**) Ctrl and *Scx* cKO

*Figure 3 continued on next page*

*Figure 3 continued*

TA muscles at 14 days post-injury were sectioned and stained with H&E. (**E**) Histogram of regenerated muscle fiber cross-sectional area from data in (**D**) (N=5 mice per group). (**F**) Histogram of average Pax7⁺ SC number per imaged field (0.08 mm²) of TA muscle sections from 5 dpi Ctrl and *Scx* cKO mice (N=5 mice per group; n=2,807 Ctrl and n=442 *Scx* cKO Pax7⁺ SCs). (**G**) After 5-ethynyl-2′-deoxyuridine (EdU) administration, Ctrl and *Scx* cKO TA muscles at 5 days post-injury were sectioned and stained for Pax7, followed by EdU reaction. Arrows indicate Pax7⁺EdU⁺ cells, whereas asterisks indicate Pax7⁺EdU⁻ cells. Nuclei were stained with DAPI. (**H**) Percentages of EdU⁺ cells within the Pax7⁺ cell population of Ctrl and *Scx* cKO, from data in **G** (N=5 mice per group; n=858 Ctrl and n=325 *Scx* cKO Pax7⁺ SCs). (**I**) Averaged Pax7⁺ SC number per image field (0.4 mm²) in Ctrl and *Scx* cKO TA muscle sections from 14d post-injury samples (N=5 mice per group; n=350 Ctrl and n=186 *Scx* cKO Pax7⁺ SCs). Data are presented with the mean ± s.d.; *p*-values are indicated. (**C, E, F, H, I**) Unpaired two-tailed Student's *t*-test was applied, Scale bars = 20 μm in (**B, D, G**).

The online version of this article includes the following figure supplement(s) for figure 3:

**Figure supplement 1.** Experimental design for recombination efficiency and staining of 5 dpi and 14 dpi muscle sections in control and *Scx* cKO group.

rounded up or appeared necrotic before disappearing. We, therefore, evaluated PCD of cultured SCs by anti-cleaved Caspase 3 and found an increased rate of PCD of *Scx* cKO SCs, relative to that of the control (*Figure 4F and G*, *Figure 4—figure supplement 1J*). Our results support that Scx acts autonomously in the SC to promote proliferation, survival, and migration.

## Scx expression by single-cell RNA-sequencing (scRNA-seq)

To determine the mechanism underlying Scx's role in the SC-lineage, we employed scRNA-seq using the 10 x Chromium platform (*Figure 5—figure supplement 1A*). For this, multiple sites of BaCl₂ injection were made to TA and gastrocnemius muscles of control and *Scx* cKO mice to induce wide-spread injury and activate as many SCs as possible (*Morton et al., 2019*). Because the published sc-RNA-seq data (*Dell'Orso et al., 2019*) indicated a widespread *Scx* expression at 2.5 dpi (*Figure 2F*), we chose this time point for investigation.

YFP-marked control and *Scx* cKO SCs at 2.5 dpi were FACS-isolated and immediately subjected to scRNA-seq (*Figure 5A*). Data were analyzed using the R package Seurat and unsupervised graph-based clustering (*Hao et al., 2021*). After filtering, 11,388 control and 12,844 *Scx* cKO cells, respectively (with ~23,000 detectable genes), were qualified for analysis. We utilized uniform manifold approximation and projection (UMAP) to display all cells in the unified dataset and performed unsupervised shared nearest neighbor (SNN) clustering to partition cells into 18 (0–17) clusters (*Figure 5—figure supplement 1B*). We annotated the cell types by examining the normalized expression level and frequency of canonical cell type-specific genes. The percentages of cells within each cluster in control and *Scx* cKO were also calculated (*Figure 5—figure supplement 1C*). Clusters 0–2 and 4–11 contained the majority of cells expressing myogenic genes. Clusters 3 and 12–17 represent nonmyogenic cell types, including various immune cells, endothelial cells, and Schwann cells (presumably due to exosomal transfer of YFP mRNA). Cluster 13 was assigned as monocytes/macrophages/platelets but expressed myogenic genes. They are likely the immunomyoblasts proposed by *Oprescu et al., 2020*. Below, we focused on myogenic clusters to investigate the defects associated with *Scx* cKO.

## sc-RNA-seq confirms *Scx* expression during regenerative myogenesis

Of the 11 myogenic cell clusters, we classified them into four categories: early activated SC, activated SC, myocyte, and mature skeletal muscle (Fig. S5B). Within the categories of early activated SC and activated SC, multiple cell clusters were included and numbered as different states. Here, numbers were arbitrarily assigned and not meant to reflect their temporal sequence. Early activated SC 1–3 were represented by clusters 4, 5, 9, and expressed varying levels of *Pax7*, *Myod1*, and *Myf5* (*Figure 5—figure supplement 1B, D, E*). We assigned clusters 0, 1, 2, and 7 as activated SC 1–4, respectively, as they expressed lower levels of *Pax7* (compared to early activated SC). Further evidencing our assignment as activated SCs, more cells in these clusters expressed *Myod1*, *Myf5*, and *Hspa1a* (*Francetic and Li, 2011*; *Senf, 2013*). Cluster 6 represented early myocytes based on increased expression of *Myog* and *Mef2a.* Cluster 8 cells expressed high levels of *Mymk*, indicating that they are competent for fusion. Cluster 11 cells expressed *Myh1* and *Acta1*, representing mature muscle cell. Cluster 10 cells were unknown myogenic cells, for they expressed very low levels of myogenic genes. Among these clusters, the level and cell percentage of *Scx* expression were very low in early activated SCs and

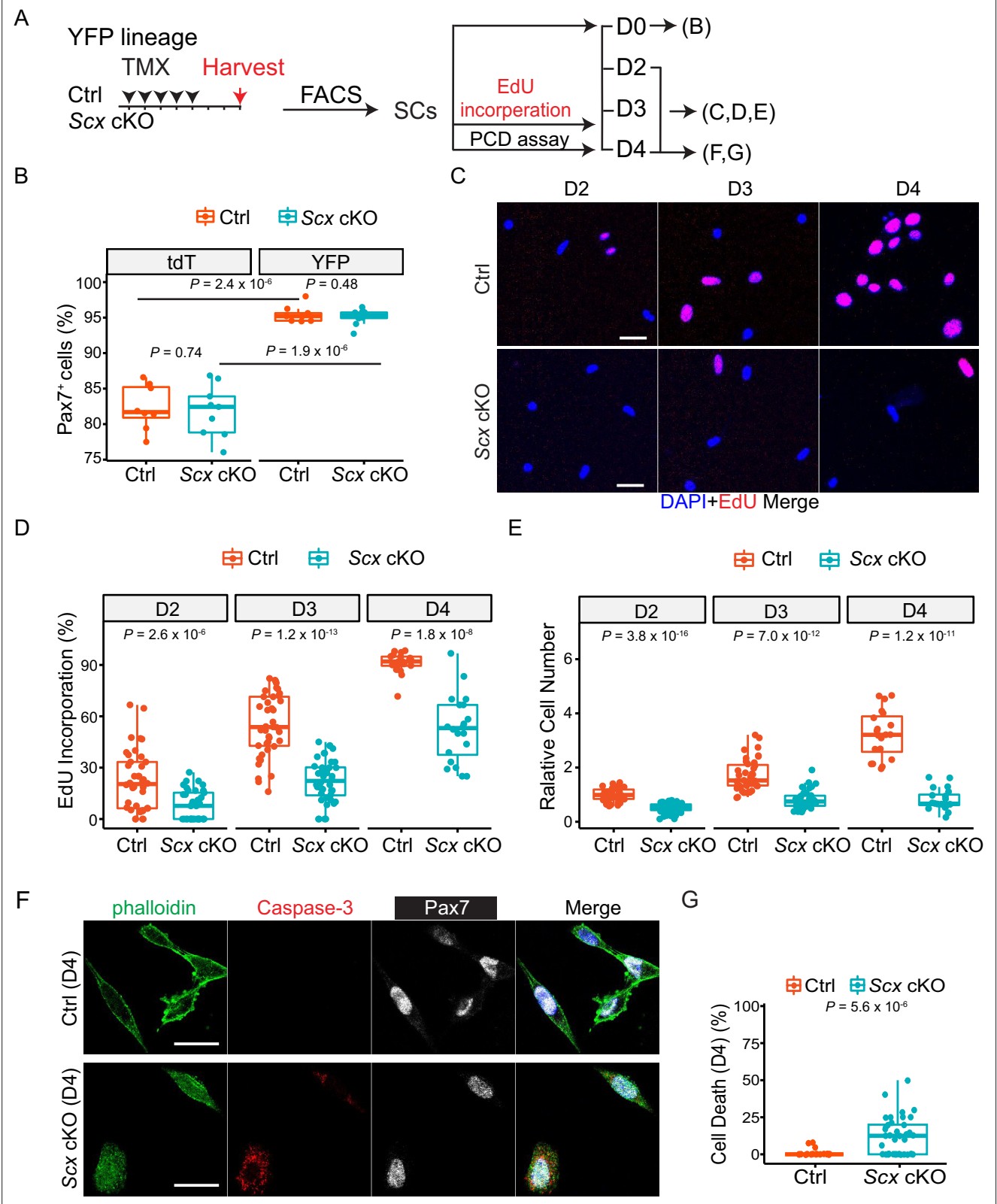

**Figure 4.** *Scx* cKO satellite cells (SCs) display a proliferation defect. (**A**) Experimental design to obtain YFP lineage-marked Pax7+ SCs for in vitro analyses in (**B-F**). (**B**) Box plot of percentages of FACS-isolated tdT and YFP marked cells expressing Pax7 by staining immediately after isolation (as D0); each dot represents one image data, 10 images per group, totally n>1000 cells for each group. (**C**) YFP lineage-marked cells were cultured in GM and assayed at days 2, 3, and 4 (D2–D4) intervals. 10 µM 5-ethynyl-2'-deoxyuridine (EdU) was added for 6 hr prior to harvesting for EdU detection. (**D**) Box

*Figure 4 continued on next page*

*Figure 4 continued*

plot of percentages of EdU⁺ cells from data in (**C**), N=2 mice, each dot represents one image data, three wells per group, eight images per well. (**E**) Box plot of ratios of total cell numbers from data in (**C**); normalized to the average control cell number at D2 as 1. (**F, G**) (**F**) FACS-isolated Ctrl and *Scx* cKO SCs were cultured in growth medium for 4 days, harvested, and immuno-stained for Pax7 and cleaved Caspase 3; actin cytoskeleton (to identify cell body) was stained by Phalloidin. (**G**) Box plot of percentages of cell death (i.e. cleaved Caspase 3⁺ cells) from data in (**F**); N=2 mice; each dot represents one image data; two wells per group, >10 images per well; 240 Ctrl and 353 *Scx* cKO cells examined. Nuclei were stained with DAPI; Scale bar = 20 µm. Data are presented with the mean ± s.d.; adjusted *p*-values are shown. (**B**) Two-way ANOVA; (**D–F**) Unpaired two-tailed Student's *t*-test.

The online version of this article includes the following figure supplement(s) for figure 4:

**Figure supplement 1.** Experimental design for live imaging, live-imaging results, and programmed cell death analyses in control and Scx cKO mice.

gradually increased from activated SCs to early myocytes, fusion-competent myocytes, and mature muscle cells (*Figure 5—figure supplement 1D, F*).

We carried out Monocle 2 trajectory analysis to depict the progression of myogenic cell clusters (*Figure 5B*). Given that *Scx* expression is very low in the early activated SC category and we observed ScxGFP only in activated SC experimentally, we excluded early activated SC 1–3 from analysis. The trajectory revealed a time line consistent with our assignment, from activated SC 1 to mature muscle cells. Of the 4 activated SC clusters, activated SC 1 cells were distributed throughout the activation time line up to early myocyte stage, activated SC 2 and SC 3 cells were preferentially located in earlier time lines, whereas activated SC 4 cells were found in a later time, revealing their different states. Early myocytes, fusion-competent myocytes, and mature muscle cells were ordered as expected.

## scRNA-seq data help identify myogenic differentiation and fusion defects

To understand the timing of *Scx* action, we compared the relative densities of various cell types/states between the control and *Scx* cKO cells along the pseudotime (*Figure 5C*). Relative to control, a higher density of *Scx* cKO cells, i.e., peak 2 in *Figure 5C*, was noted just before their reduction, i.e., peaks 3 and 4. Peaks 3 and 4 correspond to fusion-competent myocytes and mature muscle cells, respectively. This information redirected us to investigate *Scx* function in fusion and differentiation. For this, control and *Scx* cKO SCs were isolated, cultured, plated at the same density, and then switched to differentiation medium (DM) (*Figure 5D*). They were assessed for expression of MyoG (for differentiation index) and MHC (for fusion index) daily over 3 days. More control cells expressed MyoG and MHC when compared to *Scx* cKO cells at each time point (*Figure 5E–G*). At day 3, *Scx* cKO cells caught up in differentiation index (still lower than that of control cells) but were still considerably lower in fusion index. This experimental result, aided by pseudotime analysis, supports a role of *Scx* for regenerative myogenic differentiation.

## Molecular pathways governed by *Scx* in regenerative myogenesis

To gain molecular insight, we examined differentially expressed genes (DEGs) between control and *Scx* cKO cells along the pseudotime line. We were particularly intrigued by the DEGs in peak 2, as it represents an early time point of difference to capture candidate direct targets of Scx. There were 3956 DEGs in peak 2 – *Scx* exhibited the largest $\log_2$ fold change in *Scx* cKO (*Figure 5H*; *Supplementary file 1a*). In particular, cyclin-dependent kinases *Cdk1* and *Cdk2* were down-regulated, and CDK-inhibitors *Cdkn1a* and *Cdkn1c* were up-regulated in *Scx* cKO cells at, and prior to, peak 2 (*Figure 5—figure supplement 2A*). This helps explain the proliferation defects of *Scx* cKO cells. However, higher cell density with less proliferation potential is somewhat counterintuitive. We suggest that *Scx* cKO cells not only proliferate slower but also progress slower towards differentiation and fusion, resulting in their stalling and accumulation at the peak 2 transitional juncture (*Figure 5C*).

Consistent with the phenotype of *Scx* cKO, Gene Ontology (GO) term analysis of peak 2 DEGs revealed that control cells showed enrichment of up-regulated genes in the categories of muscle differentiation, growth, and development, among other pathways overlapping with cardiac muscles (*Figure 5I*). 17 genes involved in the muscle cell apoptotic process were found, consistent with increased PCD detected in vitro. Unexpectedly, *Mymk* and *Mymx*, indispensable for myocyte fusion, were expressed higher in *Scx* cKO than control cells (*Supplementary file 1a and b*), possibly reflecting their compensatory up-regulation due to compromised differentiation/fusion of *Scx* cKO

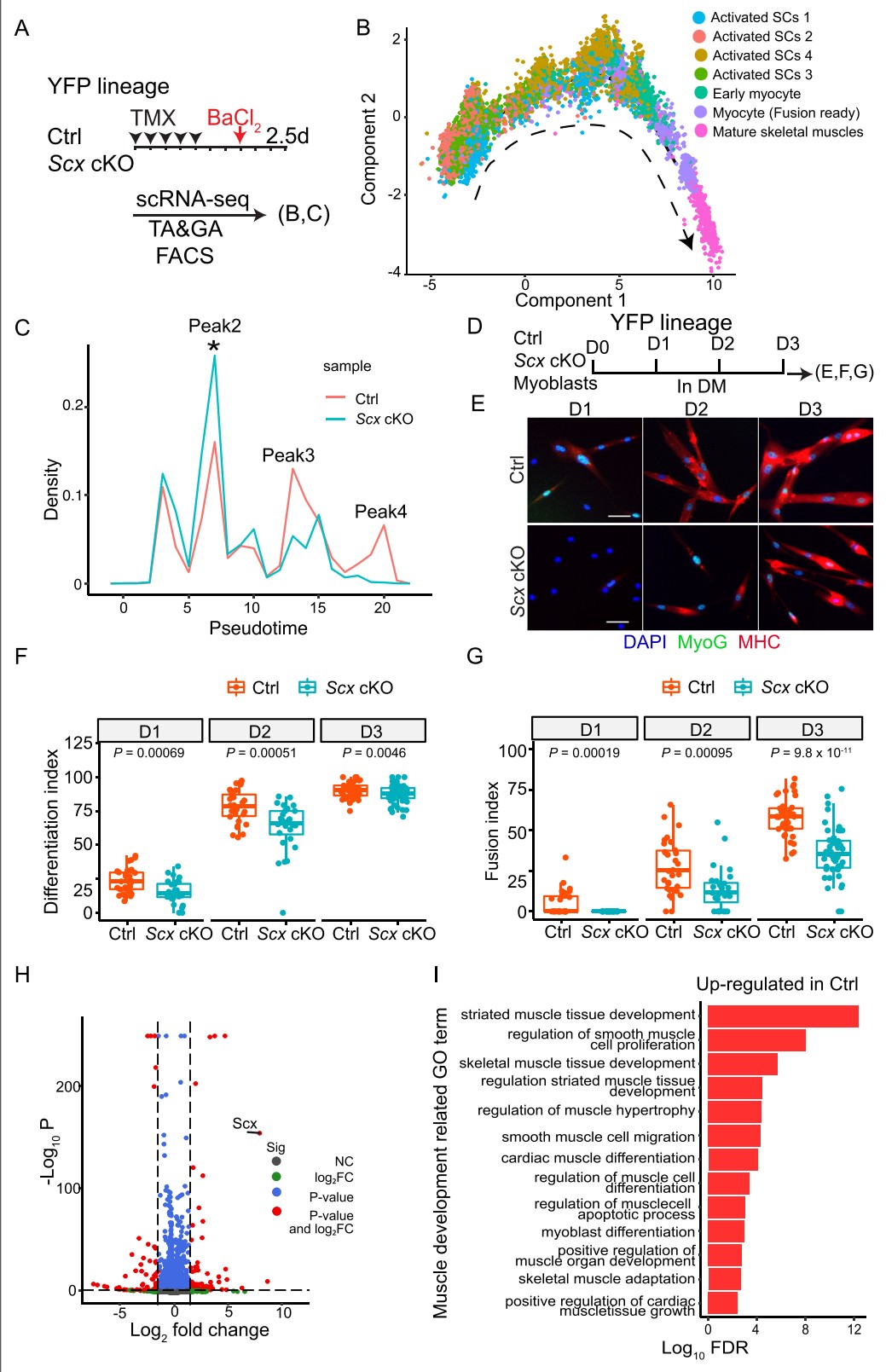

**Figure 5.** Single-cell RNA-sequencing (scRNA-seq) helps identify the role of *Scx* in myogenic differentiation and fusion. (**A**) Satellite cell (SC) scRNA-seq scheme for YFP lineage-marked SCs. YFP⁺ cells were FACS-isolated from 2.5 dpi BaCl₂ injured tibialis anterior (TA) and the gastrocnemius (GA) muscles. (**B**) Trajectory analysis of the seven myogenic clusters (complete cell cluster analysis in *Figure 5—figure supplement 1*) indicated to the

*Figure 5 continued on next page*

*Figure 5 continued*

right. Arrowindicates the direction of pseudotime trajectory. (**C**) Cell densities of Ctrl and *Scx* cKO cells along the trajectory in (**B**). Cell in Peaks 2–4 were used for the differentially expressed gene (DEG) analysis; the asterisk indicates Peak 2 as our main focus. (**D**) In vitro differentiation assay scheme. SC-derived myoblasts were cultured in GM for 12 hr (D0), switched into differentiation media (DM), and harvested daily for analysis over 3 days (D1-D3). (**E**) Myoblasts subjected to the scheme in (**D**) were stained for MyoG (for differentiation index in **F**) and for myosin heavy chain (MHC) (for fusion index in **G**). Nuclei were stained with DAPI; Scale bar = 20 µm. (**F-G**). Box plot of differentiation index (**F**) and fusion index (**G**) from data in (**E**). Each dot represents one image data. Unpaired two-tailed Student's *t*-tests were applied and adjusted *p*-values are shown. (N=3 mice; three wells per group per time point; 10 images per well; in total, 3342 control, and 2561 *Scx* cKO cells examined). (**H**) Volcano plot of relative gene expression (Log2 fold change) in Ctrl versus ScxKO cells in Peak 2 (in **C**). (**I**) Gene Ontology (GO) term enrichment of muscle development-related processes from DGEs in (**H**).

The online version of this article includes the following figure supplement(s) for figure 5:

**Figure supplement 1.** Assembly and curation of single-cell RNA-sequencing (scRNA-seq) atlas of muscle stem cell.

**Figure supplement 2.** Relative expression levels of four cell cycle genes.

cells. Analyses of peak 3 and 4 DEGs provide additional information about selective differentiation processes being disrupted in *Scx* cKO cells (see Discussion).

## Identification of direct targets by CUT&RUN assay

To uncover direct gene targets of Scx that regulate muscle differentiation and/or maturation, we utilized the CUT&RUN (*Kaya-Okur et al., 2020*) assay to determine Scx bindings sites in the genome. To aid this endeavor, a triple-Ty1 tag (3XTy1) was fused to the C-terminus of Scx to create a *Scx^Ty1* allele (*Figure 6—figure supplement 1A*). *Scx^Ty1/Ty1* mice are viable and fertile without apparent tendon abnormality. Ty1 was detected in linearly arrayed patellar tenocytes (*Figure 6—figure supplement 1B*) and in cultured myoblasts (*Figure 6—figure supplement 1C, D*) derived from *Scx^Ty1/Ty1* mice. During the differentiation time course over 3 days in culture, the largest fraction of cells with detectable Ty1 presented at day 1 (*Figure 6—figure supplement 1E*).

We performed the CUT&RUN using anti-Ty1 on *Scx^Ty1/Ty1* myoblasts (Scx-CUT&RUN) at 12 hr after switching them to DM (*Figure 6—figure supplement 1F*); this time point was chosen to uncover early targets. We included two controls: ScxGFP myoblasts with anti-Ty1 and *Scx^Ty1/Ty1* myoblasts with non-specific IgG. A total of 1003 binding peaks were identified in 861 gene loci with 33.4%, 38.88%, and 22.44% located in intergenic regions, introns, and promoters, respectively, alongside other genomic regions, respectively (*Figure 6B*; *Supplementary file 1c*). These peaks were enriched for the bHLH transcription factor binding motif, the E-box: CAG(A/C)TG (*Figure 6C*), indicating high data quality. By integrating the Scx-CUT&RUN data with DEGs in the scRNA-seq data of *Scx* cKO cells (specifically those in peak 2 of *Figure 5C*), we found 207 intersecting genes (*Figure 6D*; *Supplementary file 1d*). Scx-binding peaks at these gene loci were also enriched for the E-box motif (*Figure 6E*), implicating these genes as direct targets. As expected, GO terms of these genes showed enrichment for processes in muscle differentiation, fusion, and myofibril assembly (*Figure 6F*; *Supplementary file 1e*). We also noted the enrichment for processes of mRNA destabilization, catabolism, poly(A) shortening, etc., suggesting that mRNA metabolism is altered in the *Scx* cKO (*Figure 6F*; *Supplementary file 1e*). Four of the 207 candidate direct target genes provide possible explanations for defective differentiation of *Scx* cKO cells: *Mef2a*, *Capn2*, *Myh9*, and *Cflar* (see Discussion). The Scx-CUT&RUN peaks at these loci were in either the promoter region or intron (*Figure 6G, F*), suggesting that Scx binding and/or function is not confined to the promoter region. Taken together, Scx directly regulates a set of E-box-containing genes, and we discuss how some of these genes help explain the phenotype observed below.

## Discussion

Here, we show that *Scx* is expressed in activated mouse SCs, and it regulates many aspects of muscle regenerative process, from proliferation, cell survival, migration, to differentiation and fusion. The Scx target genes we identified underscore its function in muscle regeneration.

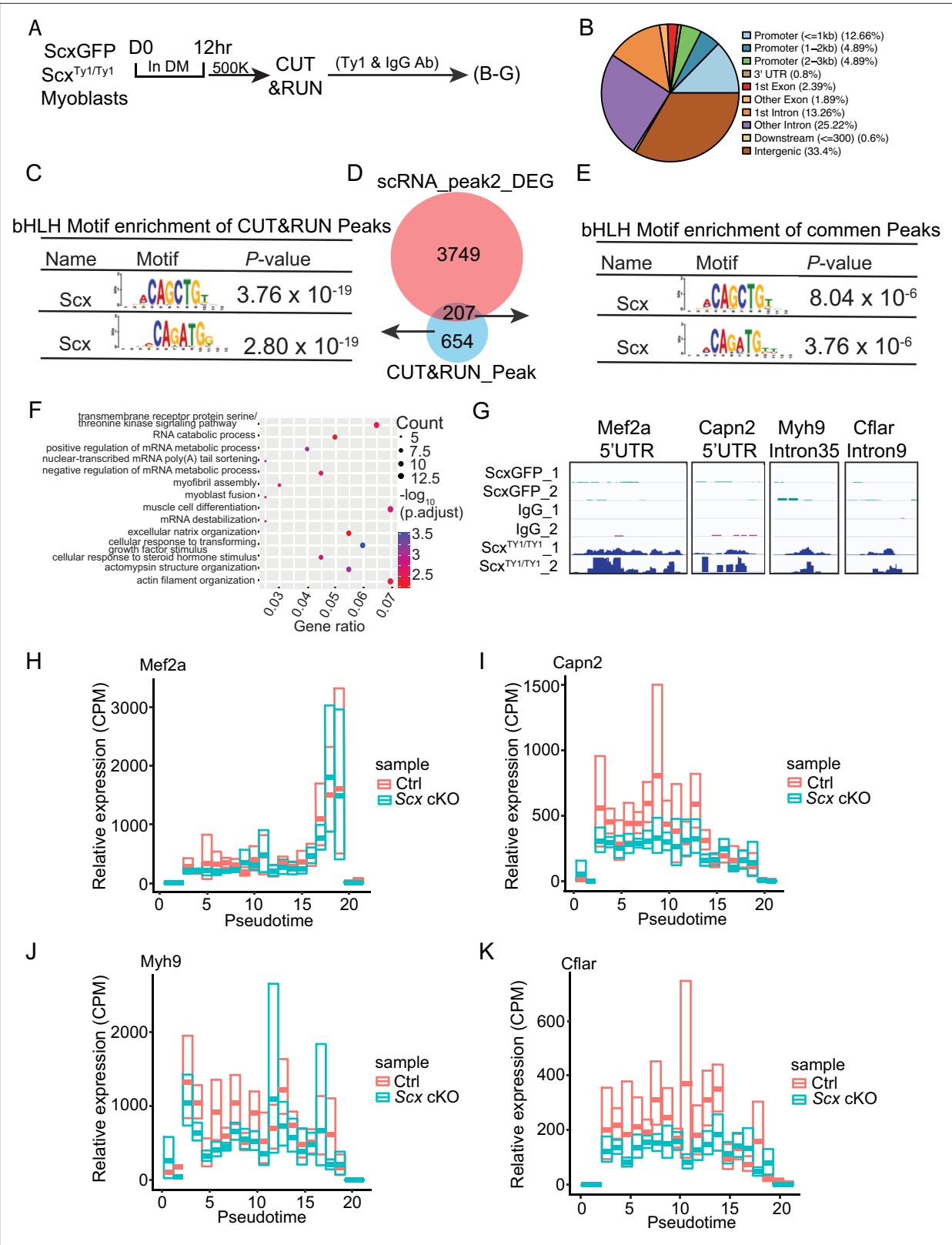

**Figure 6.** CUT&RUN and single-cell RNA-sequencing (scRNA-seq) identify direct targets of Scx. (**A**) Experimental scheme for CUT&RUN profiling of the Scx binding in the genome of Scx[Ty1/Ty1] and ScxGFP myoblasts. Primary myoblasts derived from SCs of Scx[Ty1/Ty1] (experimental group) and ScxGFP (control group) mice were used. They were cultured in GM for 12 hr (D0) and switched to differentiation media (DM) for 12 hr for use. 500,000 (500K) cells per group were subjected to CUT&RUN using an anti-Ty1 antibody or an IgG control antibody, in duplicate. (**B**) Pie chart for distribution of Scx CUT&RUN

*Figure 6 continued on next page*

*Figure 6 continued*

peaks in various regions of the genome. (**C**) Motif enrichment analysis with SEA from MEME suite (v. 5.5.0) identified bHLH protein binding motif (i.e. E-box) in all Scx CUT&RUN binding peaks. (**D**) Venn diagram of intersecting genes (207 genes) between Scx CUT&RUN target genes (861) and DEGs (3956) in Peak 2 of *Figure 5C*. (**E**) Motif enrichment analysis (as in C) of the 207 genes in (**D**) also showed enrichment of bHLH protein binding motifs, the E-box. (**F**) GO term analysis of the 207 genes in (**D**). Gene Ontology (GO) terms with *p*<0.0001 were plotted. (**G**) Genomic snapshots of Scx CUT&RUN peaks on four select genes related to muscle differentiation. (**H**) Expression levels (CPM, counts per million UMI) of the four select genes in (**G**) along the pseudotime trajectory (same trajectory as *Figure 5C*).

The online version of this article includes the following figure supplement(s) for figure 6:

**Figure supplement 1.** Ty1 expression in tendon and myoblasts, genomic snapshots of four select genes related to muscle differentiation.

## The multiplicity of Scx lineage

Since the initial description of the *Scx* gene (*Cserjesi et al., 1995*), most efforts have been focused on its role in tendon. Its early expression in the syndetome and the limb mesenchyme eventually becomes realized in tendons, ligaments, and CT (*Schweitzer et al., 2001*; *Brent et al., 2003*; *Tozer and Duprez, 2005*; *Pryce et al., 2007*). Lineage tracing by *Scx^Cre^* confirmed the aforementioned descendant cell types alongside other cell types (*Esteves de Lima et al., 2021*; *Ono et al., 2023*). Of relevance, a lineage contribution to myofibers was found. The temporal emergence of Scx[+] cells with myogenic potential was not provided by constitutive Cre-mediated lineage tracing. On the other hand, TMX-inducible lineage tracing mediated by the CT marker gene *Ors1* (i.e. using an *Ors1^CreERT2^*) revealed myogenic incorporation competence that declines towards late embryogenesis (*Esteves de Lima et al., 2021*). A Prx1[+] CT population has also been shown to incorporate into the myofiber near the myotendonous junction (MTJ) at neonatal stages (*Yaseen et al., 2021*). Consistently, scRNA-seq of embryonic chick limb mesenchyme identified a cell cluster co-expressing CT and myogenic signatures at multiple stages (*Esteves de Lima et al., 2021*). Whether these bi-potential CT/myogenic cells arise from dermomyotome, syndetome, or a yet-to-be-identified origin remains to be rigorously examined.

We show here that adult SCs express ScxGFP upon injury and culture, and that ScxGFP is co-localized with Pax7, MyoD, and MHC. scRNA-seq data confirm endogenous *Scx* expression in multiple regenerative myogenic clusters/states, in which the other CT markers *Twist2*, *Ors1*, and *Pdgfra* are barely detectable. Moreover, only the lineage-marked Scx[+] cells induced after, but not prior to, injury contribute to regenerative muscles and SCs. Together, these results support that muscle interstitial Scx[+] CT (lineage-marked prior to injury) have no myogenic potential, whereas activated Pax7[+] SCs expressing *Scx* (lineage-marked after injury) can contribute to new muscles and SCs. This is consistent with transplanted Scx[+] CT (*Giordani et al., 2019*) lacking a contribution to muscle. Adult muscle interstitial CT are highly heterogeneous within a muscle group as well as between muscle groups based on scRNA-seq data, and not all CT express *Scx* (*Muhl et al., 2020*). Anatomically, adult muscle interstitial Scx[+] cells are paramysial cells that line the perimysium (*Muhl et al., 2020*). Lineage tracing data showed that Scx[+] CT and MTJ cells were descendants of Hic1[+] MPs, but no myofiber incorporation from the Hic1[+] lineage was noted (*Scott et al., 2019*). Whether CT/myogenic bipotential progenitors exist in adult muscle is of considerable interest. Regardless, our results strongly support that SCs express *Scx* after activation and require *Scx* function for efficient regeneration.

## *Scx* function in tendon versus muscle

*Scx* has been considered a master regulator of tendon (and ligament) development as *Scx* mutant mice develop severely compromised tendons in the limbs and tail (*Murchison et al., 2007*; *Yoshimoto et al., 2017*; *Shukunami et al., 2018*). *Scx* is required for the expression of multiple tendon matrix protein-encoding genes, such as *Col1a1*, *Col3a1*, and *Tnmd* (*Shukunami et al., 2018*), but not for tendon progenitor specification. Ablation of embryonic Scx[+] cells led to mis-patterned muscle bundles (*Ono et al., 2023*), supporting an interdependence between muscle and tendon for connectivity (*Kardon, 1998*). Retrospectively, the observed muscle mispattern by ablating Scx[+] cells likely included ablation of CT/myogenic cells and tendon cells. In adults, *Scx* continues to be required for tendon growth and repair after injury (*Howell et al., 2017*; *Sakabe et al., 2018*; *Gumucio et al., 2020*; *Korcari et al., 2022*). By contrast, we focused on *Scx* function in proliferation, migration, differentiation, and fusion within the Pax7[+] SC lineage for muscle regeneration.

## Downstream genes with implications for the myogenic defect of *Scx* cKO

GO-term analyses and literature reviews of the 207 DEGs from our scRNA-seq and CUT&RUN data sets identified genes in myogenic processes, instead of genes in tenogenic or CT processes. Several of these downstream genes help us understand how *Scx* may act to regulate regenerative myogenesis: *Mef2a*, *Capn2*, *Myh9*, and *Cflar*. Knocking out and knocking down *Mef2a* led to compromised myoblast differentiation in vivo and in vitro, respectively (*Seok et al., 2011*; *Liu et al., 2014*; *Estrella et al., 2015*; *Wang et al., 2018*). Reduced *Mef2a* levels explain the compromised myoblast differentiation of *Scx* cKO cells. Consistently, several *Mef2a* target genes, such as *Hspb7*, *Atp1a2*, *Tmem182* (*Wales et al., 2014*) were also downregulated (*Supplementary file 1a and f*). Capn2 is a calpain isoform expressed in the skeletal muscle, and the locus harbors 5 E-boxes and 1 MEF-2 binding site (*Dedieu et al., 2003*). Knocking down *Capn2* in C2C12 cells led to compromised cell migration and fusion (*Honda et al., 2008*), as observed for *Scx* cKO cells. *Myh9* was shown to regulate bipolar cell morphology and alignment during myocyte fusion in vitro (*Swailes et al., 2006*). Its downregulation is consistent with the defective fusion of *Scx* cKO cells. Lastly, *Cflar* were shown to proliferation and prevent apoptosis in vascular smooth muscle cells and T lymphocytes (*Wang et al., 2002*; *Zhang and He, 2005*; *Budd et al., 2006*; *Vesely et al., 2009*). It may act similarly in the SC to explain reduced proliferation and increased cell loss of *Scx* cKO cells. These four genes displayed reduced expression levels at the early part of the pseudotime trajectory (*Figure 6H–K*), consistent with them being direct targets. The other 203 genes likely also contribute to aspects of the *Scx* cKO phenotype in ways yet to be determined. Taken together, Scx directly regulates a set of E-box containing genes, and several of these genes have direct implications to the phenotype observed.

## Scx downstream target genes in tendon versus muscle

As bHLH proteins, both Scx and Myod1 bind E-box, CANNTG; the central two nucleotides distinguish binding affinities for different bHLH proteins. The initial characterization of Scx showed that it only binds to the left E-box (CATGTG) in the enhancer (with 10 E-boxes) of the muscle creatine kinase (*MCK*) gene (*Cserjesi et al., 1995*), whereas Myod1 has a higher affinity to the right E-box (CACCTG). Not surprisingly, the right E-box (with high affinity for Myod1) is more important than the left E-box for *MCK* expression (*Nguyen et al., 2003*). By contrast, characterization of the promoter of a tendon-specific gene *Tnmd* identified two Scx-responsive E-boxes, CAGATG and CATCTG (*Shukunami et al., 2006*; *Shukunami et al., 2018*). Our CUT&RUN identified CAG(A/C)TG as high-ranking Scx binding motifs in myogenic cells, which is the same as one of the E-boxes in the *Tnmd* promoter. Recently, bulk-RNA-seq and ChIP-seq were combined to define *Scx* target genes in embryonic tenocytes (*Li et al., 2021*). Although their and our data sets are not age-matched and obtained by different methods, we compared them nonetheless. Overall, DEGs (including those without Scx-binding sites) between our and their data yielded minimal overlap (0.9%, using the criteria of log2FC >0.5). Two genes, *Htra3* and *Olfml2b*, are overlapping DEGs (48 genes for tenocytes and 207 genes for myoblasts) with Scx-binding sites, and neither gene has been studied in tendon or skeletal muscle. Importantly, the compiled E-box sequences bound by Scx in tenocytes and myoblasts are not different, i.e., CAG(A/C)TG. Thus, the deployment of *Scx* by adult SCs is not a re-use of its function in the tendon. The distinctiveness of Scx target genes between these two tissues is most likely attributed to chromatin accessibility imposed by different epigenomes.

## Indirect target genes of Scx further explain defects of the *Scx* cKO

Although we emphasized Scx's direct target genes in peak 2 of *Figure 5C*, there were many more DEGs that were indirect targets (i.e. without significant Scx-CUT&RUN peaks). Dysregulation of those genes also provides insights to *Scx*-regulated muscle regeneration. For the proliferation defect, we mentioned four dysregulated cell cycle regulators in the results section. In addition, *Erk1/2/3* (*Mapk1/3/6*), known for their role in cell growth, also exhibited lower expression levels in *Scx* CKO cells during the early pseudotime phase (*Supplementary file 1a and f*). For GO-enrichment in cell migration (20 genes; *Supplementary file 1b*), *Itga2* and *Crk* are worth noting as they have been shown to play this role in non-muscle cell contexts (*Ren et al., 2019*; *Cai et al., 2022*, *Chuang et al., 2018*, *Huang et al., 2015*). They may mediate myogenic cell migration under the umbrella program of *Scx*. For differentiation and fusion at the later pseudotime phase, there are 1942 and 755 DEGs

(*Supplementary file 1f and g*) in peaks 3 and 4 (*Figure 5C*), respectively. GO-term analyses identified 74 (in peak 3) and 47 (in peak 4) genes related to muscle differentiation (*Supplementary file 1h and i*). Several of them have documented roles in myogenic differentiation, e.g., *Hacd1* (*Lin et al., 2012*, *Blondelle et al., 2015*), *Klhl41* (*Paxton et al., 2011*; *Ramirez-Martinez et al., 2017*), *Ehd2* (*Doherty et al., 2008*; *Posey et al., 2011*), and *Lmna* (*Dubinska-Magiera et al., 2013*; *Maggi et al., 2016*). As these gene products act in different cellular compartments and mediate distinct processes, *Scx* does not appear to govern a singular process for muscle differentiation. How these indirect genes come to be dysregulated in the absence of *Scx* remains to be deciphered.

Together with the embryonic CT/myogenic bipotential cells and the Prx1⁺ CT capable of myogenic fusion near the MTJ, our results add an additional layer of complexity and further blur the molecular and cellular boundaries that divide muscle versus tendon/CT identity. The wealth of information on heterogeneous cell types and states obtained by scRNA-seq will continue to break many long-accepted concepts of tissue-restricted functions of transcription factors.

## Methods

### Mouse strains

*Pax7^CE/+^* (*Pax7^Cre-ERT2^*) (*Lepper et al., 2009*), *Rosa^YFP^* (Gt(ROSA)26^Sortm19(EYFP)Cos/J^) (*Srinivas et al., 2001*), *Rosa^tdT^* (Gt(ROSA)26^Sortm14(CAG-tdtomato)Hze/J^) (*Madisen et al., 2010*), *Scx^F^* (*Scx^tm1Stzr^*) (*Murchison et al., 2007*), *Scx^CreERT2^* (*Scx^tm2(cre/ERT2)Stzr^*) (*Howell et al., 2017*) and Tg-ScxGFP (*Pryce et al., 2007*) alleles were obtained from either original investigators or the Jackson Laboratory (JAX). *Scx^Ty1^* allele was made and characterized by our group, with 3 Ty1 tags (EVHTNQDPLD) inserted upstream of the TGA codon of the *Scx* gene. All animals had mixed backgrounds. Genotypes of animals are stipulated in text, figures, and legends. For qPCR to determine *Scx* cKO efficiency, primers are in *Supplementary file 1j* (referenced in *Figure 3—figure supplement 1* legend). Both sexes were used in all experiments and grouped together, except that only males were used for scRNA-seq. All mice were used between 2–4 months of age. All animal treatment and experiments were approved by the Institutional Animal Care and Use Committee (IACUC) of the Carnegie Institution of Washington (Permit number A3861-01).

### TMX and EdU administration

Tamoxifen (TMX; Sigma) was prepared as 20 mg ml⁻¹ stock in corn oil (Sigma) and administered by intraperitoneal injection to the mice at 4 mg per 40 g body weight following regimens in text, figures, and legends. For daily in vivo proliferation tracing, 5-ethynyl-2'-deoxyuridine (EdU, 0.5 mg/ml in PBS; Thermo Fisher Scientific) was administered by intraperitoneal injection at 0.1 mg per 20 g body weight per injection. Muscle samples were collected as specified in figures and legends.

### Muscle injury

For CTX injury, control and experimental mice were anaesthetized by isoflurane/oxygen vapor, tibialis anterior (TA) muscle was injected with 50 µl of 10 µM CTX (Cardiotoxin, Sigma-Aldrich) using an insulin syringe (U-100; BD); For BaCl₂ injury, control and experimental mice were anaesthetized with 2,2,2-tribromoethanol (Sigma) which was prepared as a 100% (w/v) stock solution in 2-methyl-2-butanol (Sigma), diluted 1:40 in PBS, This anesthetic was delivered through intraperitoneal injection at 10 µl per 1 g body weight. Muscle injury was administered by injecting 2–4 ul per site of 1.2% (w/v) barium chloride (Fisher Chemical) into approximately 25 sites in the lower hindlimb muscles. Animals were then harvested at the post-injury time point stated in the text and figure legend.

### Muscle sample preparation

TA muscle samples were collected, fixed for 8 min in ice-cold 4% paraformaldehyde (PFA) (EM Grade, cat, 157–4) in PBS, sequentially incubated in 10, 20, and 30% sucrose/PBS overnight, embedded in OCT compound (Tissue-Tek, #4583), frozen in isopentane (Sigma)/liquid nitrogen and stored at −80 °C until cryosectioning. Cross-sections (10 µM) of the mid-belly region of the muscle were stained with haematoxylin and eosin (H&E; Surgipath) or used for immunostaining and EdU reactions.

### SC isolation by FACS and myoblast culture

SCs were isolated according to the protocol described previously (*Liu et al., 2015*; *Yue et al., 2020*) with slight modifications. Briefly, mouse hindlimb muscles were dissected, minced, and digested with

collagenase II (1000 U/ml, Worthington) in wash medium (10% Horse Serum (HS, Invitrogen)) in Ham's F-10 medium with 1% penicillin/streptomycin (P/S, Gibco) for 1.5 hr followed by centrifugation and washing. Then, the tissue slurry was further digested by collagenase II (100 U/ml) and dispase (1.1 U/ml, Gibco) in wash medium for 0.5 hr to get single cell suspension for cell sorting. The cell suspension was sorted using a BD ARIA III sorter equipped with 375 nm, 488 nm, 561 nm, and 633 nm lasers. For fluorescence sorting, the YFP$^+$ (tdT$^+$) cells are sorted with green fluorescence. FITC channel 488 nm (red fluorescence; PE channel 568 nm). For four surface makers labeling, cells were incubated with 4,6-diamidino-2-phenylindole (DAPI) and fluorophore-conjugated antibodies (BioLegend) against CD31, CD45, stem cells antigen-1 (Sca1), and vascular cell adhesion protein 1 (Vcam1) at 4 °C for 0.5 hr. After washing, cells were subjected to FACS (DAPI$^-$, CD31$^-$, CD45$^-$, Sca1$^-$, and Vcam1$^+$ cells were collected), and data were collected by FACS Diva software v.6.1.3 (BD Biosciences). A small fraction of sorted cells was immunofluorescence staining for the muscle stem cell markers Pax7. For short-time cell culture, freshly sorted mononuclear SCs were plated on Matrigel (catalogue no. 354248; Corning) coated dishes (37 °C for 1 hr) and cultured in SCs culture medium (growth medium: 20% FBS, 5% horse serum, 1% penicillin/streptomycin, 1% GlutaMAX supplement (Gibco), and 2.5 ng/ml FGF (R&D systems) in DMEM (Gibco)) at 37 °C in tissue culture incubators with 5% $CO_2$. Cells were harvested as specified in the text and figure legend. For long-time cell culture to get a stable primary myoblast cell line, freshly isolated satellite cells were cultured in Ham's F10 (F10, Sigma), 10% HS, and 1% P/S for 2 days, then passage the cell into culture medium and expanded the cells for two more passages. After that, the cells were cryopreserved into liquid nitrogen for later CUT&RUN and myoblast differentiation and fusion assays. For in vitro differentiation and fusion assays, freshly sorted cells or frozen cells were thawed and cultured in growth medium for 12 h then changed into differentiation medium (2% HS, 1% P/S in DMEM) on Matrigel-coated plates and harvested as specified in the figure legend. For EdU labeling, 10 μM EdU was added to the SC culture medium for 6 hr before harvesting for assay.

## Live imaging
Freshly isolated SCs were cultured in growth medium on Matrigel-coated 48-well dish at 5 K cells per well, three wells per sample, and five locations per well. Images were collected every 10 min for 4 days. A short interval at the end of each day was used to adjust the focus and add medium to get quality video and keep the cell in a good state. The videos were collected with a Nikon Ti2 system.

## Immunofluorescence staining and detection
Muscle sections were hydrated with PBS, permeabilized with 0.5% Triton X-100 (Sigma-Aldrich)/PBS (0.5% PBT) for 15 min, washed with 0.05% PBT, and blocked with MOM block (Vector Lab) overnight. Sections were washed and incubated in blocking solution (1 X carbo-free blocking solution (Vector Lab) and 10% goat serum in 0.05% PBT) for 2 hr at room temperature, followed by incubation with primary antibodies diluted in blocking solution overnight at 4 °C. Sources and dilution for primary antibodies are provided in *Supplementary file 1k*. Sections were then washed with 0.05% PBT three times and incubated with appropriate Alexa Fluor-conjugated secondary antibodies (1:1000 for Alexa 488 and Alexa 568 and 1:500 for Alexa 647; Thermo Fisher) in blocking buffer for 1 hr at room temperature. Sections were then washed with 0.05% PBT, stained with DAPI (1 μg/ml in 0.05% PBT), and mounted in anti-fade diamond solution (Invitrogen). tdT fluorescence was preserved, so no antibody staining was used. This protocol was also used for SCs and myoblasts with two modifications: (1) the cells were fixed for 10 min in 4% PFA and (2) cells were only blocked with blocking buffer (10% goat serum in 0.05% PBT). For EdU detection, the Click-iT Reaction Kit (Thermo Fisher Scientific) was used before blocking according to the manufacturer's recommendations.

### TUNEL assay
The TUNEL assay kit (Cell Signaling #48513; Fluorescence, 594) was procured from Cell Signaling Technology, and the assay was performed according to the provided protocol with minor modifications. Briefly, pre-fixed frozen sections were rinsed and permeabilized as described in the immunofluorescence staining protocol. Subsequently, each section was incubated with TUNEL Equilibration Buffer for 5 min. Following the removal of the Equilibration Buffer, sections were immediately incubated in 50 μL of TUNEL reaction mix, prepared by adding 1 μL of TdT Enzyme to 50 μL of TUNEL Reaction Buffer, for 2.5 hr at 37 °C. The sections were then rinsed three times in PBST for 5 min each.

Samples were either mounted or further processed for immunostaining with immunofluorescence staining protocol.

## CUT&RUN

CUT&RUN experiments were carried out according to the CUTANATM CUT&RUN Protocol version 1.8 with modifications. Briefly, 500 k cells were used for each sample, and 0.01% digitonin (w/v) was used during the whole process. The antibodies used in the procedure were provided in *Supplementary file 1k*. For the library preparation and sequencing, ThruPLEX DNA-seq Kit (Takara) was used to construct the CUT&RUN DNA library for sequencing on an Illumina platform. 5–10 ng purified CUT&RUN-enriched DNA was used for the library preparation. The whole process was performed according to the protocol with deviations aiming to preserve short DNA fragments (30–80 bp). After the Library Synthesis step (adaptor ligation), 1.8 x volume of AMPure XP beads was added to the reaction to ensure high recovery efficiency of short fragments. 12 cycles of PCR amplification system was used, then the reaction was cleaned up with 1.2 x volume of AMPure XP beads. The libraries were assayed with a High Sensitivity DNA bioanalyzer (Agilent) for quality control and sequenced in the Illumina NextSeq 500. To enable determination of fragment length, paired-end sequencing was performed (2×75 bp, 8 bp index). The data is analyzed by nf-core/CUT&RUN pipeline with –seacr stringent parameters (version 2.0) (*Ewels et al., 2020*). The overlapped peaks between replicates were considered as conserved peaks and used for downstream analysis. The motif analysis is carried out with SEA from MEME suite (v. 5.5.0) to identify the enrichment of bHLH family binding motifs in CUT and RUN targets.

## Microscopy and image processing

H&E staining images of TA muscle sections were captured by a Nikon 800 microscope with X20 Plan Apo objectives and with a Canon EOS T3 camera using EOS Utility image acquisition software v.2.10. Fluorescent images of TA muscle sections and cultured myoblasts were either captured by a Nikon Eclipse E800 microscope equipped with X20/0.50 Plan Fluor, X40/0.75 Plan Fluor, and Hamamatsu C11440 digital camera using the Meta Morph Microscopy Automation and Image Analysis Software v.7.8.10.0, or captured by a Leica SP5 confocal microscope equipped with a X63/1.4 Plan Apo oil objective using the Leica Application Suite Advanced Fluorescence software version 2.7.3.9723. The same exposure time was used and the images were processed and scored in a blinded fashion using ImageJ v.64 (National Institutes of Health (NIH)). If necessary, brightness and contrast were adjusted for an entire experimental image set. Cell number, fiber diameter, fiber number, and fiber cross-sectional area were measured with ImageJ v.64.

## Single-cell RNA sequencing (scRNA-seq)

The lower hindlimb muscles (TA and gastrocnemius muscles) of the *Scx* cKO and control mice were injured with $BaCl_2$ and recovered for 2.5 day. Pax7 lineage cells were FACS-isolated by YFP fluorescence. Cells were suspended in PBS and counted by hemocytometer into 1000 cells/µl. Around 17,000 cells per sample were used for single-cell library preparation using the 10 x Genomics platform with Chromium Next GEM Single Cell 3′ GEM, Library and Gel Bead Kit v.3.1 (PN-1000121, v.3 chemistry), Single Cell 3′ A Chip Kit (PN-1000009), or Chromium Next GEM Chip G Single Cell Kit (PN-1000127), and i7 Multiplex Kit (PN-120262). We followed the 10 x protocol exactly to prepare the scRNA-seq library. In brief, for v.3 chemistry, 16.5 µl cell suspension and 26.7 µl nuclease-free water were mixed with 31.8 µl reverse transcription master mix. Of this 75 µl mix, 70 µl was loaded into the Chromium Next GEM Chip G. After barcoding, cDNA was purified and amplified with 11 PCR cycles. The amplified cDNA was further purified and subjected to fragmentation, end repair, A-tailing, adaptor ligation, and 14 cycles of sample index PCR. Libraries were sequenced using Illumina NextSeq 500 for paired-end reads.

## Analyses of scRNA-seq data

Sequencing reads were processed with the Cell Ranger version 6.0.1 (10X Genomics, Pleasanton, CA) using the mouse reference transcriptome mm10. From the gene expression matrix, the downstream analysis was carried out with R version 4.0.2 (2020-06-22). Quality control, filtering, data clustering and visualization, and the differential expression analysis were carried out using Seurat version 4.0.3 R

package (*Hao et al., 2021*). Cells with <1000 UMIs or mitochondrial reads >10% were removed from the analysis. In addition, we removed potential doublets by DoubletFinder (v. 2.0.3)(*McGinnis et al., 2019*). After log-normalizing the data, the expression of each gene was scaled regressing out the number of UMI and the percentage of mitochondrial genes expressed in each cell. The two data-sets were integrated with the IntegrateData function from Seurat. We performed PCA on the gene expression matrix and used the first 20 principal components for clustering and visualization. Unsu-pervised shared nearest neighbor (SNN) clustering was performed with a resolution of 0.6, and visu-alization was done using uniform manifold approximation and projection (UMAP). The Scx-expressed myogenic lineage clusters 0, 1, 2, 6, 7, 8, and 11 were subjected to trajectory analysis by Monocle 2 (v. 2.16.0)(*Qiu et al., 2017*). To organize cells in pseudotime, we performed new dimension reduc-tion and regressed out mitochondrial effects with the reduceDimension function and unsupervised clustered them into 5 clusters with the clusterCells function. The differentially expressed genes were calculated by differentialGeneTest, and the top 500 differentially expressed genes are used to order and then used by Monocle for clustering and ordering cells using the DDRTree method and reverse graph embedding.

## Quantification and statistical analysis

Statistical analyses were performed in R version 4.0, with tidyverse and ggplot2 packages. The statis-tical significance of results was determined by unpaired Student's t-test and two-way ANOVA.

## Acknowledgements

We thank Eugenia Dikovsky and the mouse facility crew for animal housekeeping, Allison Pinder for technical assistance in scRNA-seq, Mahmud Sidiqqi for assistance in microscopy, and L Yue for assis-tance with FACS of SCs. This work is supported by the NIH (AR060042 and AR071976) and the Carn-egie Fund to CMF.

## Additional information

### Funding

| Funder | Grant reference number | Author |
|---|---|---|
| National Institutes of Health | AR060042 | Chen-Ming Fan |
| National Institutes of Health | AR071976 | Chen-Ming Fan |

The funders had no role in study design, data collection and interpretation, or the decision to submit the work for publication.

### Author contributions

Yun Bai, Conceptualization, Data curation, Formal analysis, Validation, Investigation, Visualization, Methodology, Writing – original draft, Project administration, Writing – review and editing; Tyler Harvey, Conceptualization, Writing – review and editing; Colin Bilyou, Writing – review and editing; Minjie Hu, Data curation, Software, Visualization, Writing – review and editing; Chen-Ming Fan, Conceptualization, Supervision, Funding acquisition, Writing – review and editing

### Author ORCIDs

Yun Bai ⬚ https://orcid.org/0009-0003-8282-2381
Minjie Hu ⬚ https://orcid.org/0000-0002-8758-0567
Chen-Ming Fan ⬚ https://orcid.org/0000-0003-3211-6617

### Ethics

All animal treatment and experiments were approved by the Institutional Animal Care and Use Committee (IACUC) of the Carnegie Institution of Washington (Permit number A3861-01).

Reviewer #1 (Public review): https://doi.org/10.7554/eLife.95854.3.sa1
Reviewer #2 (Public review): https://doi.org/10.7554/eLife.95854.3.sa2
Author response https://doi.org/10.7554/eLife.95854.3.sa3

## Additional files

### Supplementary files

Supplementary file 1. Supplementary tables. (a) Different gene expression (DEGs) between control and Scx cKO group in peak 2. 3956 genes differently expressed between control and *Scx* cKO cells in peak 2 along the trajectory in *Figure 5C*. (b) Muscle development-related GO term enrichment from DGEs in peak 2. 16 GO term enrichments of muscle development-related processes from DGEs in Ctrl versus ScxKO cells in Peak 2 along the trajectory in *Figure 5C*. (c) Distribution of Scx CUT&RUN peaks. A total of 1003 binding peaks were identified in 861 gene loci, with 33.4%, 38.88%, and 22.44% located in intergenic regions, introns, and promoters, respectively, alongside other genomic regions, respectively. (d) Common Gene list between Scx CUT&RUN target genes and DEGs in Peak 2. 207 intersecting genes between Scx CUT&RUN target genes (861) and DEGs in Peak 2. (e) GO term enrichment of the 207 common genes. 67 GO term enrichment analysis of the 207 common genes. (f) Different gene expression (DEGs) between control and Scx cKO group in peak 3. 1942 genes were differently expressed between control and *Scx* cKO cells in peak 3 along the trajectory in *Figure 5C*. (g) DEGs between control and Scx cKO group in peak 3. 755 genes were differently expressed between control and *Scx* cKO cells in peak 4 along the trajectory in *Figure 5C*. (h) GO term enrichment from DGEs in peak 3. Go term analysis of the DGEs in peak 3, muscle-related processes were marked in red. (i) GO term enrichment from DGEs in peak 4. Go term analysis of the DGEs in peak 4, muscle-related processes were marked in red. (j) Primer pairs used in this study. PCR primer sets P1, P2, and P3 were used to detect exon 1 (E1), intron 1 (I1) and E2 of the *Scx* gene, respectively, PCR primer set used for CUT&RUN library preparation. (k) Antibodies information. Sources and dilution for primary and secondary antibodies are provided.

MDAR checklist

### Data availability

Mouse single-cell RNA sequencing data and CUT&RUN data were uploaded to NCBI (PRJNA1050758). Select intermediate RDS objects are available at figshare (https://doi.org/10.6084/m9.figshare.24783750).

The following datasets were generated:

| Author(s) | Year | Dataset title | Dataset URL | Database and Identifier |
|---|---|---|---|---|
| Bai Y, Harvey T, Bilyou C, Hu M, Cm Fan | 2023 | scRNA-seq on mouse muscle stem cell and CUT&RUN on mouse muscle stem cell | https://www.ncbi.nlm.nih.gov/bioproject/PRJNA1050758/ | NCBI BioProject, PRJNA1050758 |
| Bai Y | 2024 | Scx project related RDS files | https://doi.org/10.6084/m9.figshare.24783750 | figshare, 10.6084/m9.figshare.24783750 |

The following previously published datasets were used:

| Author(s) | Year | Dataset title | Dataset URL | Database and Identifier |
|---|---|---|---|---|
| Dell'Orso S, Juan AH, Ko K, Naz F, Gutierrez-Cruz G, Feng X, Sartorelli V | 2019 | Single-cell analysis of homeostatic and regenerative adult skeletal muscle stem cells | https://www.ncbi.nlm.nih.gov/geo/query/acc.cgi?acc=GSE126834 | NCBI Gene Expression Omnibus, GSE126834 |

*Continued on next page*

*Continued*

| Author(s) | Year | Dataset title | Dataset URL | Database and Identifier |
|---|---|---|---|---|
| De Micheli AJ, Laurilliard EJ, Heinke CL, Ravichandran H, Paula F, Sharon SB, Olivier E, Benjamin DC | 2020 | Single-cell transcriptomic atlas of FACS-sorted mouse muscle tissue cells | https://www.ncbi.nlm.nih.gov/geo/query/acc.cgi?acc=GSE143435 | NCBI Gene Expression Omnibus, GSE143435 |
| Michelle R, Liangji L, Chen-Ming F | 2019 | Targeting β1-integrin signaling enhances regeneration in aged and dystrophic muscle in mice | https://www.ncbi.nlm.nih.gov/sra/?term=SRP070128 | NCBI Sequence Read Archive, SRP070128 |
| Madaro L, Torcinaro A, De Bardi M, Contino FF, Pelizzola M, Diaferia RG, Imeneo G, Bouchè M, Puri PL, De Santa F | 2019 | Expression profiles of satellite cells and alpha7Sca1 cells in mdxITGAM-DTR mice | https://www.ncbi.nlm.nih.gov/geo/query/acc.cgi?acc=GSE134770 | NCBI Gene Expression Omnibus, GSE134770 |

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
