## [Editor Report · eLife Assessment]

This manuscript presents **important** finding regarding the regulation of a key stem cell population, namely muscle stem cells (or "satellite cells"). The evidence presented is **convincing** that Scx, a marker for tendon, is expressed in some myogenic cells and is essential for adult muscle regeneration.

---

## [Referee Report · Reviewer #1 (Public review)]

This manuscript by Bai et al concerns the expression of Scleraxis (Scx) by muscle satellite cells (SCs) and the role of that gene in regenerative myogenesis. The authors report the expression of this gene associated with tendon development in satellite cells. Genetic deletion of Scx in SCs impairs muscle regeneration, and the authors provide evidence that SCs deficient in Scx are impaired in terms of population growth and cellular differentiation. Overall, this report provides evidence of the role of this gene, unexpectedly, in SC function and adult regenerative myogenesis.

There are a few points of concern.

(1) From the data in Figure 1, it appears that all of the SCs, assessed both in vitro and in vivo, express Scx. The authors refer to a scRNA-seq dataset from their lab and one report from mdx mouse muscle that also reveal this unexpected gene expression pattern. Has this been observed in many other scRNA-seq datasets? If not, it would be important to discuss potential explanations as to why this has not been reported previously.

(2) A major point of the paper, as illustrated in Fig. 3, is that Scx-neg SCs fail to produce normal myofibers and renewed SCs following injury/regeneration. They mention in the text that there was no increased PCD by Caspase staining at 5 DPI. A failure of cell survival during the process of SC activation, proliferation, and cell fate determination (differentiation versus self-renewal) would explain most of the in vivo data. As such, this conclusion that would seem to warrant a more detailed analysis in terms of at least one or two other time points and an independent method for detecting dead/dying cells (the in vitro data in Fig. 4F is also based on assessment of activated Caspase to assess cell death). The in vitro data presented later in Fig. S4G,H do suggest an increase in cell loss during proliferative expansion of Scx-neg SCs. To what extent does cell loss (by whatever mechanism of cell death) explain both the in vivo findings of impaired regeneration and even the in vitro studies showing slower population expansion in the absence of Scx?

(3) I'm not sure I understand the description of the data or the conclusions in the section titled "Basement membrane-myofiber interaction in control and Scx cKO mice". Is there something specific to the regeneration from Scx-neg myogenic progenitors, or would these findings be expected in any experimental condition in which myogenesis was significantly delayed, with much smaller fibers in the experimental group at 5 DPI?

(4) The data presented in Fig. 4B showing differences in the purity of SC populations isolated by FACS depending on the reporter used are interesting and important for the field. The authors offer the explanation of exosomal transfer of Tdt from SCs to non-SCs. The data are consistent with this explanation, but no data are presented to support this. Are there any other explanations that the authors have considered and that could be readily tested?

(5) The Cut&Run data of Fig. 6 certainly provide evidence of direct Scx targets, especially since the authors used a novel knock-in strain for analyses. The enrichment of E-box motifs provides support for the 207 intersecting genes (scRNA-seq and Cut&Run) being direct targets. However, the rationale elaborated in the final paragraph of the Results section proposing how 4 of these genes account for the phenotypes on the Scx-neg cells and tissues is just speculation, however reasonable. These are not data, and these considerations would be more appropriate in the Discussion in the absence of any validation studies.

Comments on revisions:

The authors have adequately addressed all of the concerns I raised regarding the original submission. I have no further issues to be addressed.

---

## [Referee Report · Reviewer #2 (Public review)]

Summary:

Scx is a well-established marker for tenocytes, but the expression in myogenic-lineage cells was unexplored. In this study, the authors performed lineage-trace and scRNA-seq analyses and demonstrated that Scx is expressed in activated SCs. Further, the authors showed that Scx is essential for muscle regeneration using conditional KO mice and identified the target genes of Scx in myogenic cells, which differ from those of tendons.

Strengths:

Sometimes, lineage-trace experiments cause mis-expression and do not reflect the endogenous expression of the target gene. In this study, the authors carefully analyzed the unexpected expression of Scx in myogenic cells using some mouse lines and scRNA-seq data.

Weaknesses:

Scx protein expression has not been verified.

Comments on revisions:

The authors sincerely addressed all concerns, excluding the protein expression of Scx. There is convincing evidence from other experiments that indirectly indicate the protein expression of Scx. In addition, the importance of this study is solid. So, this reviewer doesn't require the authors to make more revisions.

---

## [Author Response]

The following is the authors’ response to the original reviews.

**Public Reviews:**

**Reviewer #1 (Public Review):**
This manuscript by Bai et al concerns the expression of Scleraxis (Scx) by muscle satellite cells (SCs) and the role of that gene in regenerative myogenesis. The authors report the expression of this gene associated with tendon development in satellite cells. Genetic deletion of Scx in SCs impairs muscle regeneration, and the authors provide evidence that SCs deficient in Scx are impaired in terms of population growth and cellular differentiation. Overall, this report provides evidence of the role of this gene, unexpectedly, in SC function and adult regenerative myogenesis.

We appreciate the comments and thank her/him for the support.

There are a few minor points of concern.(1) From the data in Figure 1, it appears that all of the SCs, assessed both in vitro and in vivo, express Scx. The authors refer to a scRNA-seq dataset from their lab and one report from mdx mouse muscle that also reveals this unexpected gene expression pattern. Has this been observed in many other scRNA-seq datasets? If not, it would be important to discuss potential explanations as to why this has not been reported previously.

Thanks for this question regarding data in Fig.1. We did initially use immunofluorescence staining of Pax7 and GFP on muscle sections and primary myoblast cultures prepared from Tg-ScxGFP mice to conclude that Scx was expressed in satellite cells (SCs). In addition to the cited mdx RNA-seq data, we have included a re-analysis of a published scRNA-seq data set in Fig.2E (Dell'Orso et al., Development, 2019), and our own scRNA-seq data (Fig.S5D, F). We have now re-examined an additional scRNA-seq data set of TA muscles at various regeneration time points (De Micheli et al., Cell Rep. 2020), in which Scx expression was detected in MuSC progenitors and mature muscle cells. We have added the De Micheli et al. reference and the re-analysis of that scRNA-seq data set for Scx expression as an additional panel in Fig. 2E, with accompanying text (p. 7, ln. 4-6). Thus, our immunostaining results are consistent with scRNA-seq data from our and two other independent scRNA-seq data sets.

We think that Scx expression in the adult myogenic lineage was not previously reported mainly because its expression level was low, and might be dismissed as spurious detection. Additionally, detecting such low expression levels requires sophisticated detection methods with high capture efficiency. Previous studies have noted limitations in transcript capture or transcription factor dropout in 10x Genomics-based datasets (Lambert et al., Cell, 2018; Pokhilko et al., Genome Res., 2021). The most likely and straightforward reason is that Scx was simply not a focus in prior studies amid so many other genes of interest. We have now added this last explanation in the text (p.7, ln. 8-9), following the re-analyses of Scx expression in published scRNA-seq data sets.

(2) A major point of the paper, as illustrated in Fig. 3, is that Scx-neg SCs fail to produce normal myofibers and renewed SCs following injury/regeneration. They mention in the text that there was no increased PCD by Caspase staining at 5 DPI. A failure of cell survival during the process of SC activation, proliferation, and cell fate determination (differentiation versus self-renewal) would explain most of the in vivo data. As such, this conclusion would seem to warrant a more detailed analysis in terms of at least one or two other time points and an independent method for detecting dead/dying cells (the in vitro data in Fig. 4F is also based on an assessment of activated Caspase to assess cell death). The in vitro data presented later in Fig. S4G, H do suggest an increase in cell loss during proliferative expansion of Scx-neg SCs. To what extent does cell loss (by whatever mechanism of cell death) explain both the in vivo findings of impaired regeneration and even the in vitro studies showing slower population expansion in the absence of Scx?

We appreciate these constructive suggestions. Based on the number of available control and cKO animals, we were limited to one additional time point at 3 dpi to assess PCD by TUNEL in vivo. We were disappointed again to find no appreciable levels of PCD at 3 dpi by TUNEL (new Fig.S4I), thus no quantifications were included. We also re-did the in vitro experiment using purified SCs and monitored PCD by staining for cleaved Caspase-3 using a validated tube of antibodies (positive staining after 6 h of treatment by 1 mM staurosporine of control and ScxcKO cells; included as new Fig. S4J and legend). We were pleased to find an increase of cleaved Caspase3 stained cells, i.e. PCD, of Scx-cKO SCs at day 4 in culture, compared to that of the control. We have now replaced the old Fig. 4F with new Fig.4F and 4G to document PCD. We also provided new text/legend for these new data (p.10. ln. 2-10; new legend for Fig. 4F and 4G).

(3) I'm not sure I understand the description of the data or the conclusions in the section titled "Basement membrane-myofiber interaction in control and Scx cKO mice". Is there something specific to the regeneration from Scx-neg myogenic progenitors, or would these findings be expected in any experimental condition in which myogenesis was significantly delayed, with much smaller fibers in the experimental group at 5 DPI?

We very much appreciate this comment. We agree that there is unlikely anything specific about the regeneration from Scx-negative myogenic progenitors. Unfilled or empty ghost fibers (basement membrane remnant) are expected due to small fiber and poor regeneration in the ScxcKO mice at 5 dpi. We have removed the subtitle and changed the content to an expected consequence rather than something special (p. 8, ln. 19-22).

(4) The data presented in Fig. 4B showing differences in the purity of SC populations isolated by FACS depending on the reporter used are interesting and important for the field. The authors offer the explanation of exosomal transfer of Tdt from SCs to non-SCs. The data are consistent with this explanation, but no data are presented to support this. Are there any other explanations that the authors have considered and that could be readily tested?

Thanks for highlighting this phenomenon. We struggled with the SC purity issue for a long time. The project started with using the R26RtdT reporter for tdT’s paraformaldehyde resistant strong fluorescence (fixation) to aid visualization in vivo. Later, when we used the tdT signal to purify SCs by FACS, we found that only 80% sorted tdT+ cells are Pax7+. We then switched to the R26RYFP reporter, from which we achieved much higher purity (95%) of SCs (Pax7+) by FACS. As such, we also repeated and confirmed many in vivo experimental results using the R26RYFP reporter (included in the manuscript). Due to the low purity of tdT+SCs by FACS, we discontinued that mouse colony after we confirmed the superior utility of the R26RYFP reporter for SC isolation.

We sincerely apologize for not being able to conduct further testable experiments on this intriguing phenomenon. However, this issue has since been addressed and published by Murach et al., iScience, (2021). Like our experience, they found non-satellite mononuclear cells with tdT fluorescence after TMX treatment when SCs were isolated via FACS. To determine this was not due to off-target recombination or a technical artifact from tissue processing, they conducted extensive analyses. They found that the tdT+ mononuclear cells included fibrogenic cells (fibroblasts and FAPs), immune cells/macrophages, and endothelial cells. Additionally, they confirmed the significant potential of extracellular vesicle (EV)-mediated cargo transfer, which facilitates the transfer of full-length tdT transcript from lineage-marked Pax7+ cells to those mononuclear cells. We have modified the text to emphasize and acknowledge their contribution to this important point, and explained the difference between YFP and tdT reporter alleles in more detail (p.9, ln. 11-17).

(5) The Cut&Run data of Fig. 6 certainly provide evidence of direct Scx targets, especially since the authors used a novel knock-in strain for analyses. The enrichment of E-box motifs provides support for the 207 intersecting genes (scRNA-seq and Cut&Run) being direct targets. However, the rationale elaborated in the final paragraph of the Results section proposing how 4 of these genes account for the phenotypes on the Scx-neg cells and tissues is just speculation, however reasonable. These are not data, and these considerations would be more appropriate in the Discussion in the absence of any validation studies.

We agree with this comment and have moved speculations into the Discussion (p. 15, ln. 4-15, and from p. 18, ln. 4 to p. 19, ln. 4).

**Reviewer #2 (Public Review):**
Summary:Scx is a well-established marker for tenocytes, but the expression in myogenic-lineage cells was unexplored. In this study, the authors performed lineage-trace and scRNA-seq analyses and demonstrated that Scx is expressed in activated SCs. Further, the authors showed that Scx is essential for muscle regeneration using conditional KO mice and identified the target genes of Scx in myogenic cells, which differ from those of tendons.Strengths:Sometimes, lineage-trace experiments cause mis-expression and do not reflect the endogenous expression of the target gene. In this study, the authors carefully analyzed the unexpected expression of Scx in myogenic cells using some mouse lines and scRNA-seq data.

We appreciate the comments and thank her/him for noting the strengths of our manuscript.

Weaknesses:Scx protein expression has not been verified.

We are aware of this weakness. We had previously used Western blotting (WB) using cultured SCs from control and ScxcKO mice, but did not detect endogenous Scx protein even in the control. In response to this comment, we have re-done several WB experiments using new lysates from control and ScxcKO SCs and two commercial antibodies: anti-Scx antibody 1 from Abcam (ab58655) and anti-Scx antibody 2 from Invitrogen (PA5-23943). These antibodies have been reported to detect endogenous Scx protein in tendon cells in Spang et al., BMC Musculoskelet Disord (2016) and Bochon et al., Int J Stem Cells (2021). Despite our best efforts, we were not able to detect a reliable Scx band. We have also conducted immunofluorescence using these two antibodies. Still, we failed to detect a difference of staining signals between control and cKO SCs using these antibodies. Lastly, we conducted immunofluorescence using the ScxTy1 myoblasts and we did not find the staining signal coinciding with the Ty1 signal (by double staining). We have been very frustrated by not knowing what caused this technical difficulty in our hands. Given that these were negative data, we did not include them. However, we do hope that the combined data from scRNA-seq, ScxCreERT2 lineage-tracing, Tg-ScxGFP expression, and ScxTy1 knock-in together are deemed sufficient to make up for the deficiency of data for endogenous Scx protein in regenerative myogenic cells.

**Response to Recommendations for the Authors:**

**Reviewer #1 (Recommendations For The Authors):**
p. 8: The text refers to Fig. 3I, but this should be Fig. 3H.

We apologize for the confusion. Please note that by keeping all 14 dpi data in the same row, we placed Fig.3I at an unconventional/unexpected position, i.e., next to 3D &3E, and above 3F-H. We were aware that this unconventional placement could cause confusion, and it did. With that said, we have now re-arranged the subfigures (same data content) so that the updated Fig.3 contains subfigures in the expected and proper spatial order. We double-checked the figure referral in the text (p. 8, ln. 16-17) and the text is correct – just that the original Fig.3I should have been at the original Fig.3H position and that is now corrected.

**Reviewer #2 (Recommendations For The Authors):**
(1) Given that Scx binds to the E-box and regulates gene expression, it is of interest to know the relevance between MyoD and Scx. If possible, the reviewer recommends to include some discussions.

Thanks for the comment. MyoD1 is a well-known transcript factor regulating myogenesis, whereas Scx is primarily studied in tenocytes and other connective tissues. We agree that our new findings deserve a discussion regarding the relevance between MyoD1 and Scx. We have added a description of their differences in the discussion and two new references (p.19, ln. 7-17).

(2) Considering that Scx is a transcriptional factor, it is interesting that Scx-GFP was not detected in the nuclei of regenerated myofibers. Could the subcellular localization of Scx-GFP provide some insights into the function of Scx as a transcription factor during muscle regeneration?

Tg-ScxGFP is a transgenic line generated by random insertion into the genome (Pryce et al., 2007; cited). The plasmid used for transgenesis was constructed by replacing most of Scx’s first exon with GFP, and including ~ 9Kb flanking regulatory sequences. As such, the ScxGFP is not a fusion gene, but rather that the GFP expression is regulated by Scx promoter and enhancer(s). This GFP reporter lacks a nuclear localization signal (NLS), hence it is mainly detected in the cytoplasm; some nuclear signal is detected, presumably due to GFP’s small size permitting passive diffusion into the nucleus. Thus, the GFP signal is used as a reporter for Scx expression, but GFP subcellular localization does not provide insight into Scx function per se. Conversely, ScxTy1/Ty1 is a knock-in allele created by fusing a triple-Ty1 tag (3XTy1) to the C-terminus of Scx, and we observed that Ty1 is located in the nucleus by the immunofluorescent staining. We used the Ty1 epitope to carry out CUT&RUN experiments to gain insight to the function of Scx as a transcription factor.

(3) Fig1D The number of arrows in the Merge image is not matched with others. In addition, the star mark in the Pax7 image is likely an error.

Apologies. We have now corrected these errors in the revised Fig.1D.

(4) FigS1A Is there only one myofiber shown in the dashed line in this image? It is unclear why only this myofiber is surrounded by the dashed line.

The dashed line encircles a single fiber because it was not visible in the provided image. However, there are 3 fibers in this image. Because we did not immuno-stain for myofibers here, we circled one fiber for illustration. For clarity, we brightened the background (of the entire original images) so the background signals from myofiber boundaries are discernable without outlines.

(5) FigS1B There was no overlapped DAPI staining in the Myogenin+ cell. DAPI-staining should be present in Myogenin+ cells because myogenin is located in the nucleus.

Fig.S1B is immuno-staining for MyoD , and we marked one MyoD+DAPI+GFP+ cell/nucleus. Fig.S1C is immune-staining for Myogenin, and we also marked one (cell/nucleus) that is triple positive.

(6) The position of the asterisk for the ScxGFP in FigS1D is misaligned. In addition, the position is not matched with Fig1C. Because all myofibers are Scx-positive, it is strange that only one myofiber has an asterisk. The reviewer suggests removing the mark.

Thank you for pointing out these errors. We have now corrected the misalignment and removed the unnecessary asterisk.